# Impact of liver-specific survival motor neuron (SMN) depletion on central nervous system and peripheral tissue pathology

Monique Marylin Alves de Almeida[1,2], Yves De Repentigny[1], Sabrina Gagnon[1], Emma R Sutton[1,2], Rashmi Kothary[1,2,3,4]*

[1]Regenerative Medicine Program, Ottawa Hospital Research Institute, Ottawa, Canada; [2]Centre for Neuromuscular Disease, University of Ottawa, Ottawa, Canada; [3]Department of Cellular and Molecular Medicine, University of Ottawa, Ottawa, Canada; [4]Department of Medicine, University of Ottawa, Ottawa, Canada

## eLife Assessment

This work presents an **important** mouse model for a liver-specific depletion of the Survival Motor Neuron (SMN) protein, where the liver retains 30% of functional full-length SMN protein. The authors provide a profile of phenotypic changes in liver-specific SMN depleted mice with **convincing** evidence supporting their claims.

*For correspondence:
rkothary@ohri.ca

Competing interest: The authors declare that no competing interests exist.

**Abstract** Spinal muscular atrophy (SMA) is caused by mutations in the Survival Motor Neuron 1 (*SMN1*) gene. While traditionally viewed as a motor neuron disorder, there is involvement of various peripheral organs in SMA. Notably, fatty liver has been observed in SMA mouse models and SMA patients. Nevertheless, it remains unclear whether intrinsic depletion of SMN protein in the liver contributes to pathology in the peripheral or central nervous systems. To address this, we developed a mouse model with a liver-specific depletion of SMN by utilizing an *Alb-Cre* transgene together with one *Smn*[2B] allele and one *Smn1* exon 7 allele flanked by loxP sites. Initially, we evaluated phenotypic changes in these mice at postnatal day 19 (P19), when the severe model of SMA, the *Smn*[2B/-] mice, exhibit many symptoms of the disease. The liver-specific SMN depletion does not induce motor neuron death, neuromuscular pathology or muscle atrophy, characteristics typically observed in the *Smn*[2B/-] mouse at P19. However, mild liver steatosis was observed, although no changes in liver function were detected. Notably, pancreatic alterations resembled that of *Smn*[2B/-] mice, with a decrease in insulin-producing β-cells and an increase in glucagon-producingα-cells, accompanied by a reduction in blood glucose and an increase in plasma glucagon and glucagon-like peptide (GLP-1). These changes were transient, as mice at P60 exhibited recovery of liver and pancreatic function. While the mosaic pattern of the Cre-mediated excision precludes definitive conclusions regarding the contribution of liver-specific SMN depletion to overall tissue pathology, our findings highlight an intricate connection between liver function and pancreatic abnormalities in SMA.

## Introduction

Insufficient levels of Survival Motor Neuron (SMN) protein, primarily arising from mutations or deletions in the *SMN1* gene, are the root cause of spinal muscular atrophy (SMA), a hereditary neuromuscular disorder widely acknowledged as one of the leading genetic causes of infant mortality (*Lefebvre*

*et al., 1995*; *Lefebvre et al., 1997*; *Crawford and Pardo, 1996*). Characterized by the loss of motor neurons, SMA presents as a progressive weakening of muscle strength, primarily impacting proximal muscles. If left untreated, this deterioration culminates in respiratory failure and premature mortality (*Finkel et al., 2016*).

The diverse phenotypes observed in SMA patients can be attributed to the genetic variations within the *SMN* genes (*Lefebvre et al., 1995*). Humans harbor two copies of the *SMN* gene, *SMN1* and *SMN2*, which share a nearly identical sequence except for a critical C to T substitution at position 6 of exon 7 in *SMN2* (*Lorson et al., 1999*). This single nucleotide change disrupts splicing of the gene, resulting in the predominant production of an unstable protein lacking exon 7, known as *SMNΔ7* (*Lorson et al., 1999*). Despite this, approximately 10% of functional full-length (FL) SMN protein is still produced by the *SMN2* gene (*Lefebvre et al., 1997*). Consequently, the *SMN2* gene serves as a modifier of disease phenotype, with a higher copy number correlating with milder disease severity (*Feldkötter et al., 2002*). In contrast, mice possess a single copy of the *Smn1* gene (*Lefebvre et al., 1997*). Notably, while homozygous deletion of both *SMN* genes has not been observed in humans, the homozygous deletion of the *Smn1* gene in mice induces morphological alterations and degenerative changes in embryos post-morula stage, ultimately resulting in death at preimplantation stage (*Schrank et al., 1997*). This underscores the pivotal role of the *SMN* gene in early embryonic development (*Vitte et al., 2004*).

Recent research has unveiled the extensive impact of SMN depletion beyond motor neurons, implicating additional tissues such as the liver and pancreas in both SMA patients and mouse models of SMA (*Bowerman et al., 2014*; *Bowerman et al., 2012a*; *Bowerman et al., 2012b*; *Deguise et al., 2019a*). Despite major therapeutic advancements focusing on restoring SMN levels, which have notably enhanced both lifespan and quality of life in SMA patients, complete phenotypic rescue remains elusive (*Finkel et al., 2016*; *Chiriboga, 2017*; *McMillan et al., 2022*). Additionally, existing therapies primarily target postnatal treatment, potentially overlooking crucial developmental changes stemming from SMN depletion. This underscores the likelihood of a broader role for the *SMN* gene in multiple non-neuronal cell types, necessitating further exploration.

Attention has been increasingly drawn to the liver as a critical area of investigation in SMA research. Both clinical studies and preclinical mouse models have unveiled disruptions in fatty acid metabolism, leading to an increased susceptibility among SMA patients to dyslipidemia and liver steatosis (*Deguise et al., 2019a*; *Crawford et al., 1999*; *Deguise et al., 2021a*; *Deguise et al., 2019b*; *Deguise et al., 2021b*). Given its multifaceted role in regulating numerous biological functions, including the storage and regulation of lipids, carbohydrates, amino acids, and iron, as well as the synthesis of essential growth factors and the clearance of toxic metabolites, any disruptions in liver function may have deleterious effects on other tissues (*Trefts et al., 2017*).

To explore the role of SMN in the liver, prior studies have utilized a transgenic mouse model featuring the Cre recombinase transgene driven by the α-fetoprotein promoter (*Alfp-Cre*). This approach led to near-total depletion of SMN protein specifically in the liver, resulting in severe impairment of liver development and late embryonic lethality (*Vitte et al., 2004*). In our study, to better represent the level of SMN depletion in the liver observed in SMA and elucidate the intricate interplay between SMN deficiency in the liver and its systemic impact in SMA-like pathology, we utilized a mouse model employing *Alb-Cre*, where the albumin promoter drives expression of Cre recombinase in a liver-specific manner. This model also harbors one $Smn^{2B}$ allele and one *Smn1* exon 7 allele flanked by loxP sites (F7). The $Smn^{2B}$ allele contains a 3-base pair substitution in the mouse *Smn1* gene, resulting in aberrant splicing of exon 7 (*DiDonato et al., 2001*; *Hammond et al., 2010*). Consequently, this genetic approach results in the production of approximately 30% full-length functional SMN protein production in the liver.

Our findings reveal that our novel liver-specific SMN depleted (herein referred to as $Alb^{Cre/+}$;$Smn^{2B/F7}$) mouse model induces mild liver steatosis and disrupts pancreatic function at postnatal day 19 (P19), characterized by decreased insulin-positive and increased glucagon-positive cells, alongside reduced blood glucose levels. However, both liver and pancreatic defects were rescued by P60. Remarkably, this mouse model does not exhibit an overt SMA phenotype, demonstrating normal lifespan and motor function. Nevertheless, our data suggests an intricate relationship between hepatic function and pancreatic abnormalities in SMA, thereby enhancing our comprehension of the disease's pathophysiology.

## Results

### Generation of liver-specific SMN-depleted mice

To elucidate the impact of liver-specific SMN depletion on both CNS and peripheral tissue pathology, we utilized mice expressing Cre recombinase under the control of the mouse albumin promoter, referred to as $Alb^{Cre}$. This strain is commonly employed for liver-specific gene knockout using the Cre/loxP system (**Postic et al., 1999**). We generated the $Smn^{2B/F7}$ genotype by crossing $Smn^{F7/F7}$ with $Smn^{2B/2B}$ mice (**Figure 1A**). The $Smn^{2B/F7}$ genotype carries one allele harboring a three nucleotide switch in the exon splicing enhancer within exon 7 of the mouse $Smn1$ gene (known as $Smn^{2B}$ allele), mimicking the $SMN2$ gene in humans (**Hammond et al., 2010**), while the other allele features exon 7 flanked by loxP sites (**Frugier et al., 2000**). Additionally, crosses between $Alb^{Cre}$ and $Smn^{F7/F7}$ yielded $Alb^{Cre/+};Smn^{F7/+}$ offspring.

The liver-specific mouse line was subsequentially established through further genetic crosses between $Alb^{Cre/+};Smn^{F7/+}$ and $Smn^{2B/F7}$, resulting in various genotypes, including $+/+;Smn^{2B/F7}$ (**Figure 1B**), and $Alb^{Cre/+};Smn^{2B/F7}$ (**Figure 1C**). As $+/+;Smn^{2B/F7}$ mice lack Cre recombinase, they were used as heterozygous controls. In contrast, recombination occurs in $Alb^{Cre/+};Smn^{2B/F7}$ animals resulting in the specific deletion of one allele of exon 7 in the liver, while the $Smn^{2B}$ allele remains unaffected.

$Smn^{2B/-}$ mice, the severe mouse model of SMA, express approximately 15% residual full-length SMN (FLSMN) compared to $Smn^{2B/+}$ heterozygous mice (**Eshraghi et al., 2016**). Therefore, both genotypes were included in this study as additional controls. Tissues were collected at postnatal day 19 (P19), corresponding to the symptomatic phase of the $Smn^{2B/-}$ mouse model. At this stage, $Smn^{2B/-}$ mice show increased liver steatosis and metabolic defects, ultimately culminating in multiple system defects and mortality, typically occurring around P25 (**Reilly et al., 2024**). In contrast, $Smn^{2B/+}$ mice have slightly reduced levels of FLSMN and no overt phenotypical changes compared to wild-type mice (**Bowerman et al., 2012a**).

We first assessed SMN protein levels in the liver and across CNS and peripheral tissues. Our findings confirmed a reduction in SMN protein levels across various tissues in $Smn^{2B/-}$ mice (**Figure 1D–H**). However, in $Alb^{Cre/+};Smn^{2B/F7}$, there was a ~70% decrease in FLSMN protein expression in the liver (**Figure 1D and D'**) compared to littermate control counterparts ($+/+;Smn^{2B/F7}$). Conversely, SMN protein levels in the brain (**Figure 1E and E'**), muscle (**Figure 1F and F'**), spinal cord (**Figure 1G and G'**) and pancreas (**Figure 1H and H'**) remained unaltered, indicating that deletion of $Smn$ exon 7 was restricted to the liver in $Alb^{Cre/+};Smn^{2B/F7}$ mice. We validated these findings by assessing Cre expression in liver, pancreas, and spinal cord tissues (**Figure 1—figure supplement 1A**). Cre expression was exclusively observed in the liver, where it exhibited variable expression across hepatocytes, with some cells showing detectable expression while others showed no expression, confirming the stochastic nature of Cre activity in our model. No Cre signal was detected in either the pancreas or spinal cord tissues.

### Alb^Cre/+^;Smn^2B/F7^ mice show mild yet important liver steatosis

We previously showed that $Smn^{2B/-}$ mice exhibit defects in fatty acid metabolism and lipid accumulation in the liver (**Deguise et al., 2019a**; **Deguise et al., 2021a**; **Deguise et al., 2019b**; **Deguise et al., 2021b**). Additionally, SMA patients face an elevated risk of dyslipidemia and steatosis (**Deguise et al., 2019a**). Hence, we explored whether the intrinsic decrease in SMN levels restricted to the liver would lead to increased lipid accumulation. Histology from $Alb^{Cre/+};Smn^{2B/F7}$ mice (**Figure 2A**) qualitatively indicates some level of hepatic microvesicular steatosis and lipid accumulation. However, there was a noticeable variation in cell morphology and Oil Red-O retention among hepatocytes, suggesting incomplete recombination.

Consistent with the histological findings, $Alb^{Cre/+};Smn^{2B/F7}$ mice showed variable triglycerides levels (n1=45.95, n2=10.10; n3=75.72; n4=21.91 µg/mg, mean ± SEM 38.44±14.48) (**Figure 2B**), and yet higher than those observed in $+/+;Smn^{2B/F7}$ controls (n1=2.60, n2=0.15; n3=0.66; n4=1.44 µg/mg, mean ± SEM 1.213±0.5332), although this difference did not reach statistical significance (p=0.0824). As expected, $Smn^{2B/-}$ controls exhibited increased lipid accumulation and triglyceride levels. Taken together, the data suggest an intrinsic role of SMN in regulating fat liver metabolism.

We next sought to identify whether $Alb^{Cre/+};Smn^{2B/F7}$ mice display disturbed liver homeostatic function. Specifically, we evaluated parameters associated with iron metabolism (**Figure 2C, C', D and D'**), endoplasmic reticulum stress (**Figure 2E and E'**), and insulin-like growth factor-1 (IGF-1; **Figure 2F**)

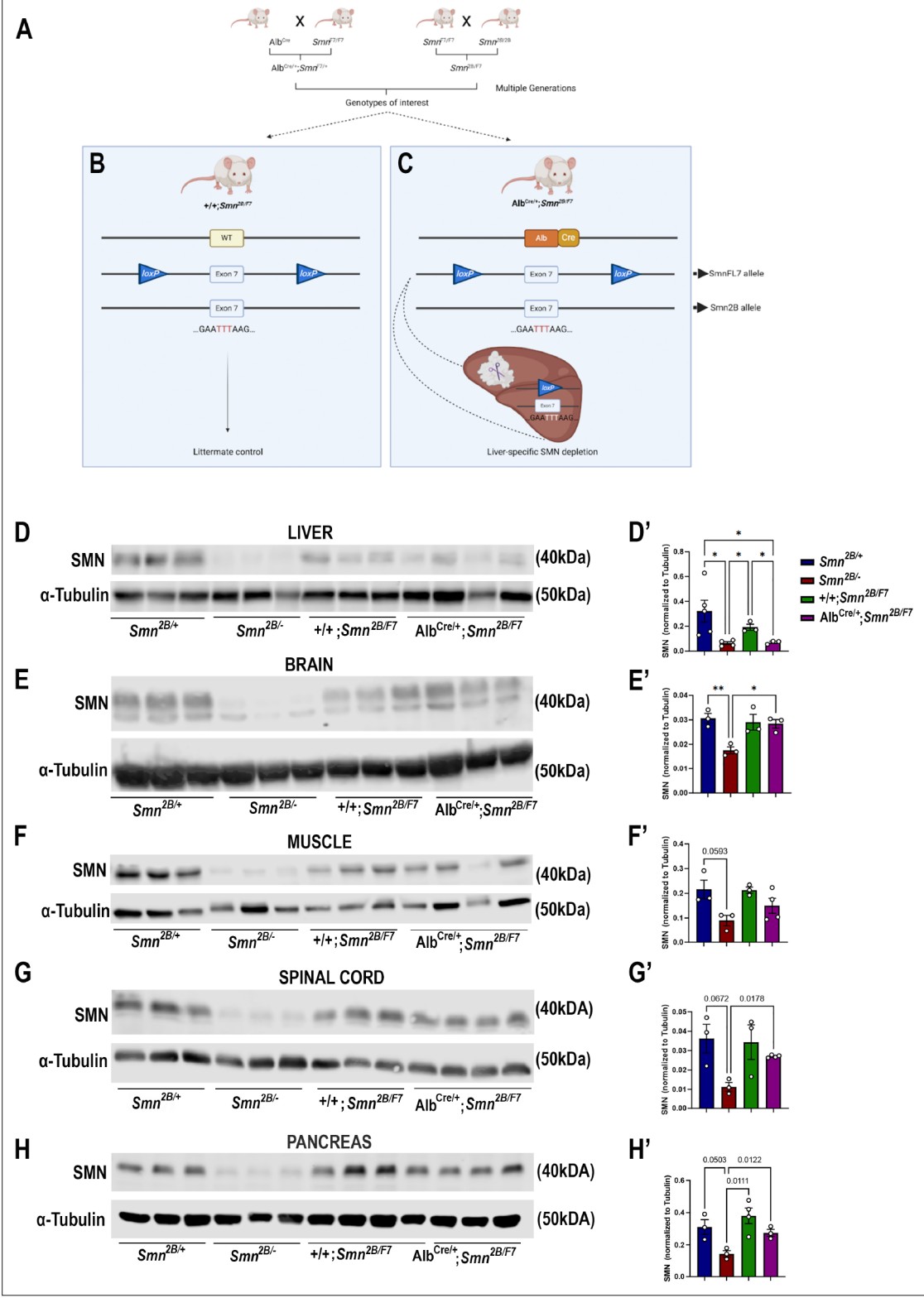

**Figure 1.** Generation of liver-specific SMN-depleted mice. (**A**) Schematic representation of the breeding scheme to generate liver-specific SMN depletion. In these mice, the *Smn1* gene carries one *Smn2B* allele that harbors a three-nucleotide switch in exon 7, and one *Smn1* allele whose exon 7 is floxed by loxP sites (*SmnF7*). The *Alb-Cre* transgene has also been crossed into this background to provide liver-specific expression of the Cre recombinase. (**B**) Littermate controls do not harbor Cre recombinase but do carry one *Smn2B* allele and are thus used as heterozygous controls. (**C**) Upon crossing *AlbCre/+;SmnF7/+* with *Smn2B/F7* animals, Cre recombination will induce specific excision of exon 7 in the liver, leading to reduced SMN protein

*Figure 1 continued on next page*

*Figure 1 continued*

production only in the liver. (**D–H**) Immunoblots were performed to assess SMN protein levels in various tissues from $Smn^{2B/+}$, $Smn^{2B/-}$, $+/+;Smn^{2B/F7}$, and $Alb^{Cre/+};Smn^{2B/F7}$ mice at P19. Membranes were probed for SMN and then reprobed for α-tubulin (loading control). (**D'-H'**) Bar graphs show quantification of SMN protein levels in the liver (**D'**), brain (**E'**), muscle (**F'**), spinal cord (**G'**) and pancreas (**H'**), normalized to α-tubulin. n ≥ 3, mean ± SEM. Statistical significance indicated by *p < 0.05, **p < 0.01 or p values, following Brown-Forsythe and Welch ANOVA.

The online version of this article includes the following source data and figure supplement(s) for figure 1:

**Source data 1.** PDF file containing original western blots for *Figure 1D–H*, indicating the relevant bands and treatments.

**Source data 2.** Original files for western blot analysis displayed in *Figure 1D–H*.

**Source data 3.** Raw data for SMN protein levels plotted in *Figure 1D'–H'*.

**Figure supplement 1.** Cre expression across different tissues.

levels, all of which have previously been identified as altered in $Smn^{2B/-}$ mice (*Deguise et al., 2021b*) and were further confirmed in our study. Our analysis revealed no significant differences in the protein levels of heme oxygenase, transferrin, P62, and IGF-1 between the $Alb^{Cre/+};Smn^{2B/F7}$ and $+/+;Smn^{2B/F7}$ control groups. These findings suggest that liver-specific SMN depletion may not play a significant role in these pathways or that the mosaic nature of Cre-mediated excision led to a subtle effect that is difficult to detect.

## Liver-specific SMN-depleted mice display pancreatic defects

We have demonstrated pancreatic and glucose abnormalities in the $Smn^{2B/-}$ mouse model of SMA and type I SMA patients (*Bowerman et al., 2012b*; *Reilly et al., 2022*). Therefore, we conducted immunohistochemistry to investigate the cellular profile of the pancreas, aiming to ascertain whether the targeted depletion of SMN in the liver contributes to pancreatic dysfunction. Surprisingly, double-labeling of insulin-producing β cells and glucagon-producing α cells revealed a significant decrease in the proportion of β cells and a notable increase in α cells in the pancreas of $Alb^{Cre/+};Smn^{2B/F7}$ mice compared to $+/+;Smn^{2B/F7}$ and $Smn^{2B/+}$ heterozygous controls (*Figure 3A, B and F*). This loss of β cells and rise in α cells could suggest a functional impairment in glucose metabolism. Consequently, we evaluated non-fasting glucose levels in these mice (*Figure 3C*), revealing reduced blood glucose levels in $Alb^{Cre/+};Smn^{2B/F7}$ mice relative to $+/+;Smn^{2B/F7}$ and $Smn^{2B/+}$. However, blood glucose levels remained higher than those observed in $Smn^{2B/-}$ mice. Overall, these findings indicate imbalances in α and β cell fate within pancreatic islets in $Alb^{Cre/+};Smn^{2B/F7}$ mice, resembling those observed in $Smn^{2B/-}$ mice.

In previous work, we demonstrated that $Smn^{2B/-}$ mice exhibit a normoinsulinemic plasma state, despite persistent hypoglycemia and reduced C-peptide levels at P19 (*Deguise et al., 2021a*), findings that were recapitulated in this study (*Figure 3C–E*). Similarly, $Alb^{Cre/+};Smn^{2B/F7}$ mice showed no significant changes in plasma insulin levels. However, surprisingly, C-peptide levels in these mice were comparable to those observed in $+/+;Smn^{2B/F7}$ and $Smn^{2B/+}$ heterozygous controls (*Figure 3E*). Additionally, we observed a marked increase in plasma glucagon levels (*Figure 3G*) as well as elevated glucagon-like peptide (GLP-1) levels, a product of proglucagon processing (*Figure 3H*), reaching levels similar to those seen in $Smn^{2B/-}$ mice. These changes likely reflect an increase in pancreatic α-cell numbers, although they were not accompanied by dysregulation of plasma pancreatic polypeptide (PP, *Figure 3I*) or amylin (*Figure 3J*) hormone production. Furthermore, the maintenance of SMN protein expression at levels comparable to those of heterozygous controls in the pancreas (*Figure 1H and H'*) suggests that selective SMN depletion in the liver may be implicated in the dysfunction of the pancreas-liver axis.

## Motor neurons, neuromuscular junction (NMJ) or motor fibers are unaffected in liver-specific SMN-depleted mice

The hallmark features of SMA include the degeneration of lower spinal motor neurons, neuromuscular junction (NMJ) pathology, and muscle atrophy (*Bowerman et al., 2012a*; *Reilly et al., 2024*; *Deguise et al., 2016*; *Deguise et al., 2020*; *Gavrilina et al., 2008*; *Murray et al., 2013*; *Murray et al., 2015*; *Shababi et al., 2014*; *Lorson et al., 2010*). While the exact mechanisms linking these changes in the motor unit to the clinical manifestations of SMA remain unclear, diminished levels of SMN protein are recognized as a pivotal factor (*Lefebvre et al., 1995*; *Lefebvre et al., 1997*; *Reilly et al., 2024*; *Lorson et al., 2010*; *Hua et al., 2011*; *Mercuri et al., 2022*). Thus, we investigated the impact of

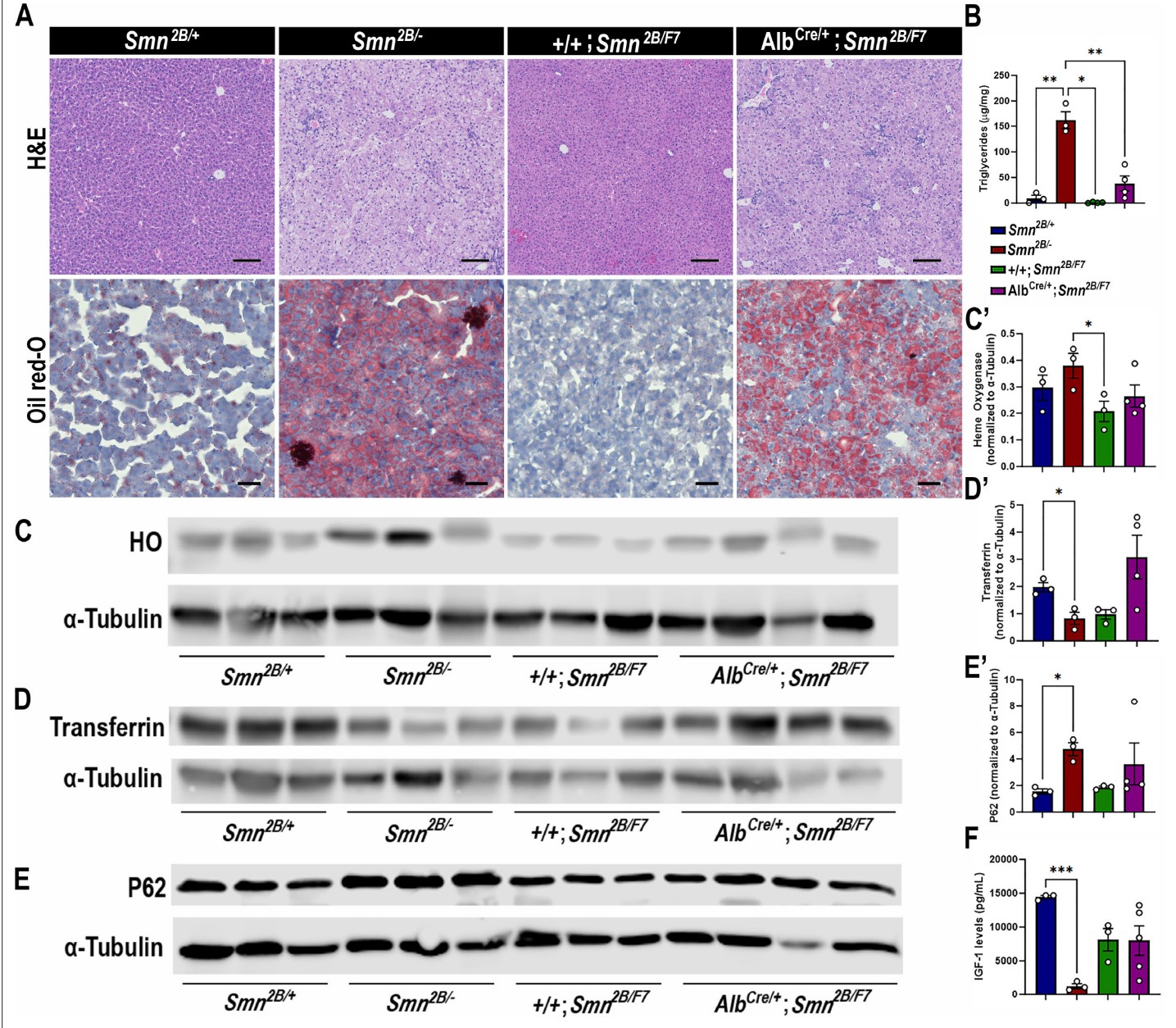

**Figure 2.** Impact of liver-specific SMN depletion in liver function. (**A**) Representative images of H&E (top row, scale bar 100 μm) and Oil Red-O (bottom row, scale bar 50 μm) stained liver sections from $Smn^{2B/+}$, $Smn^{2B/-}$, $+/+;Smn^{2B/F7}$, and $Alb^{Cre/+};Smn^{2B/F7}$ mice at P19. (**B**) Bar graph shows quantification of liver triglycerides. (**C–E**) Immunoblots were performed to assess heme oxygenase (HO, (**C-C'**)), transferrin (**D-D'**) and P62 (**E-E'**) protein levels in the liver. α-tubulin was used as a loading control. (**C'-E'**) Bar graphs show quantification of HO (**C'**), transferrin (**D'**), and P62 (**E'**) levels. (**F**) Bar graph depicts quantification of liver IGF-1. n ≥ 3, mean ± SEM. Statistical significance indicated by *p < 0.05, **p < 0.01, ***p < 0.001, following Brown-Forsythe and Welch ANOVA.

The online version of this article includes the following source data for figure 2:

**Source data 1.** PDF file containing original western blots for *Figure 2C–E*, indicating the relevant bands and treatments.

**Source data 2.** Riginal files for western blot analysis displayed in *Figure 2C–E*.

**Source data 3.** Raw data for plots in *Figure 2B, C', D', E' and F*.

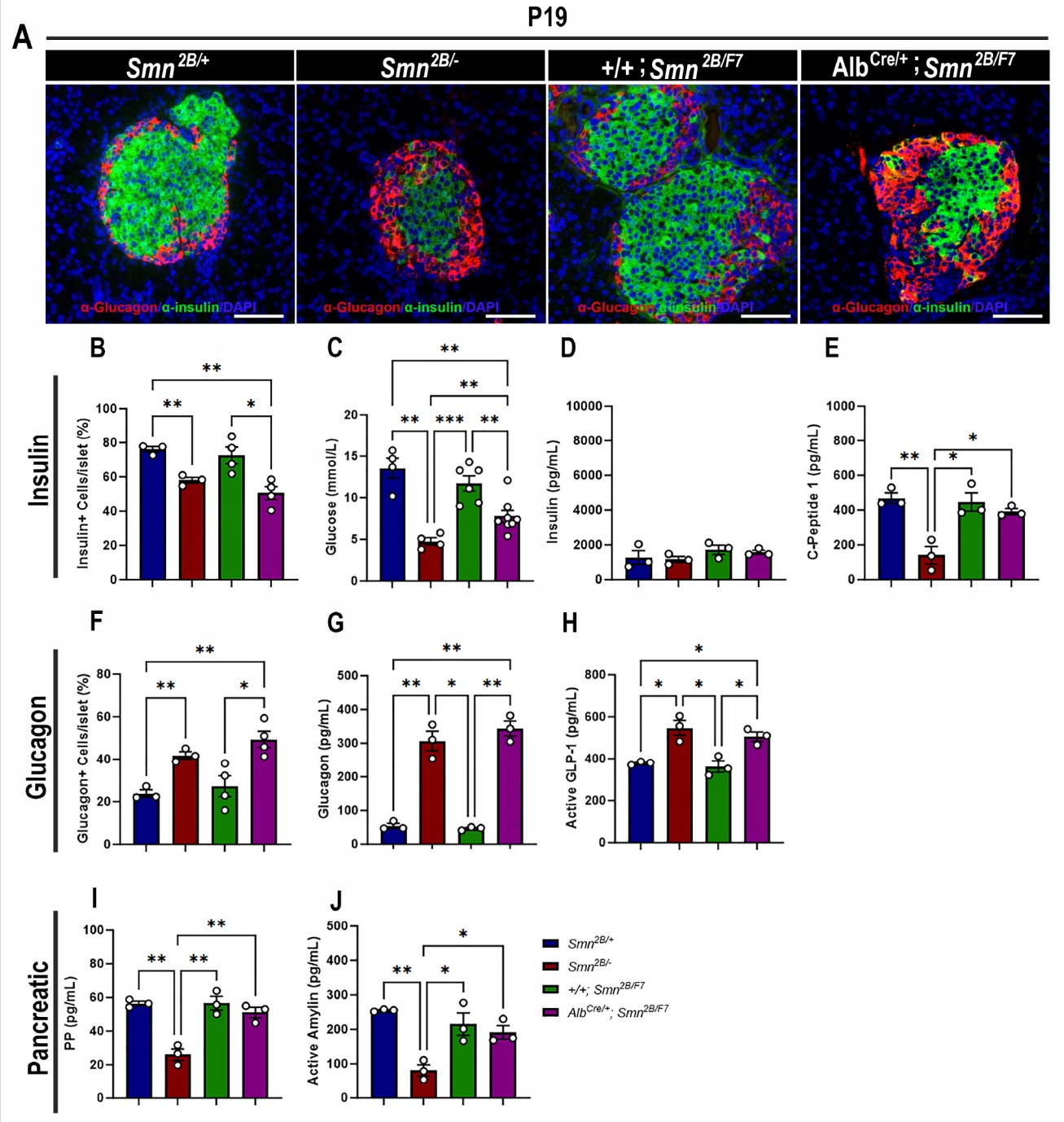

**Figure 3.** Contribution of liver-specific SMN depletion to pancreatic pathology. (**A**) Representative immunofluorescent images of pancreatic islets stained for glucagon (red) and insulin (green) from $Smn^{2B/+}$, $Smn^{2B/-}$, $+/+;Smn^{2B/F7}$, and $Alb^{Cre/+};Smn^{2B/F7}$ mice at P19. Scale bar 50 µm. (**B**) Bar graph shows quantification of insulin-positive cells relative to the total number of DAPI-positive cells within the pancreatic islet. (**C**) Bar graph depicts non-fasting blood glucose levels across different genotypes. (**D, E**) Bar graphs illustrate non-fasting plasma levels of insulin (**D**) and C-peptide (**E**). (**F**) Bar graph shows quantification of glucagon-positive cells relative to the total number of DAPI-positive cells within the pancreatic islet. (**G–J**) Bar graphs depict non-fasting plasma levels of glucagon (**G**), active GLP-1 (**H**), pancreatic polypeptide (**I**) and active amylin (**J**). n ≥ 3, mean ± SEM. Statistical significance indicated by *$p < 0.05$, **$p < 0.01$, ***$p < 0.001$, following Brown-Forsythe and Welch ANOVA.

The online version of this article includes the following source data for figure 3:

**Source data 1.** Raw data for plots in *Figure 3B–J*.

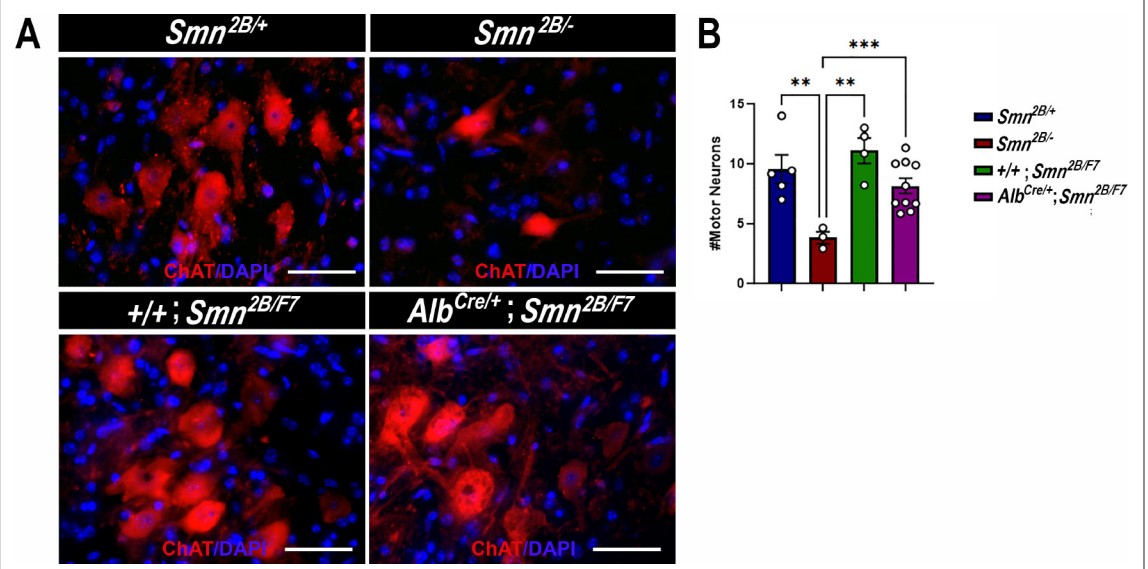

**Figure 4.** Impact of liver-specific SMN depletion on motor neuron cell body numbers. (**A**) Representative images of lumbar spinal cord anterior horns stained for ChAT (red) and DAPI (blue) from $Smn^{2B/+}$, $Smn^{2B/-}$, $+/+;Smn^{2B/F7}$, and $Alb^{Cre/+};Smn^{2B/F7}$ mice at P19. (**B**) Bar graph shows quantification of motor neuron cell body numbers. n ≥ 3, mean ± SEM. Statistical significance indicated by **p < 0.01, ***p < 0.001, following Brown-Forsythe and Welch ANOVA. Scale bar = 50 μm.

The online version of this article includes the following source data for figure 4:

**Source data 1.** Raw data for motor neuron counts in the plots in **Figure 4B**.

liver-specific SMN depletion on these SMA-like features. Our findings show no significant differences in motor neuron cell body counts (**Figure 4A and B**), neurofilament accumulation at NMJs (**Figure 5A and B**), NMJ endplate occupancy (**Figure 5A and C**), muscle fiber size (**Figure 6A and B**), or muscle fiber frequency distribution (**Figure 6C**) among $Alb^{Cre/+};Smn^{2B/F7}$ and $+/+;Smn^{2B/F7}$ and $Smn^{2B/+}$ heterozygous controls. However, as anticipated, $Smn^{2B/-}$ mice exhibited motor neuron loss, NMJ defects, and muscle atrophy.

## Liver-specific SMN-depleted mice show transitory liver and pancreatic pathology

We next assessed $Alb^{Cre/+};Smn^{2B/F7}$ mice at P60 to determine whether liver and pancreatic dysfunction persisted or improved during early adulthood. Histological analysis revealed no significant changes in hepatocyte morphology (as assessed by H&E staining) or lipid accumulation (evaluated with Oil Red O staining) in $Alb^{Cre/+};Smn^{2B/F7}$ mice at this time point (**Figure 7A**). These findings suggest that proliferating non-recombined cells may help restore liver homeostasis over time. Notably, the expression of Cre recombinase was also observed at P60 (**Figure 1—figure supplement 1B**), indicating stable Cre activity across postnatal development.

Further examination of pancreatic tissue at P60 showed no significant differences in insulin-producing β cells or glucagon-producing α cells in $Alb^{Cre/+};Smn^{2B/F7}$ mice compared to $+/+;Smn^{2B/F7}$ and $Smn^{2B/+}$ heterozygous controls (**Figure 7B, C and G**). Additionally, non-fasting plasma glucose, insulin, and C-peptide levels in $Alb^{Cre/+};Smn^{2B/F7}$ mice were similar to those in the control groups (**Figure 7D–F**). Moreover, no significant changes in plasma glucagon (**Figure 7H**), GLP-1 (**Figure 7I**), pancreatic polypeptide (**Figure 7J**), or amylin hormone production were observed. Collectively, these results suggest that the liver and pancreatic defects observed in $Alb^{Cre/+};Smn^{2B/F7}$ mice may be transient, potentially linked to the stochastic nature of Cre recombinase activity in our model and the highly proliferative properties of hepatocytes. In addition, these findings suggest that liver-specific SMN may play a role in regulating pancreatic function, a mechanism that warrants further investigation.

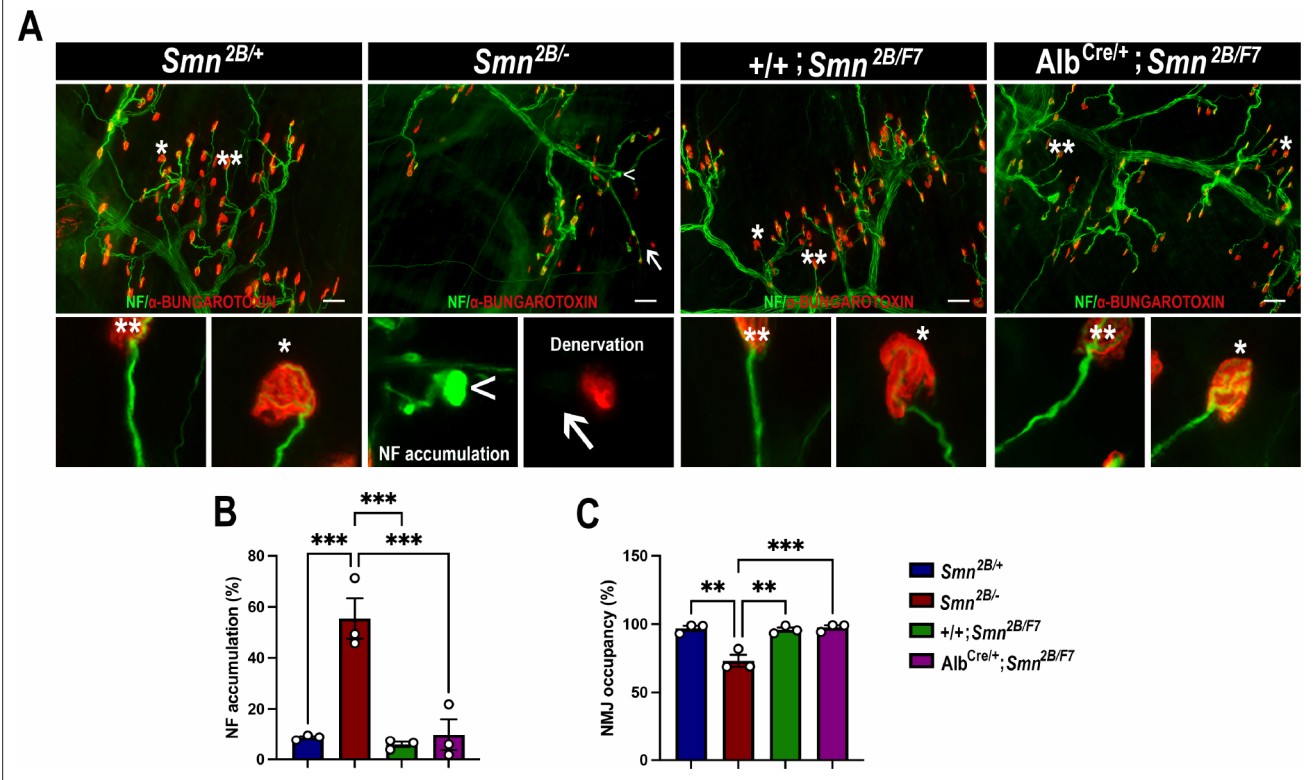

**Figure 5.** Impact of liver-specific SMN depletion on neuromuscular junction pathology. (**A**) Representative images of *transversus abdominis* (TVA) muscle stained with bungarotoxin (red), neurofilament (NF) (green) and synaptic vesicle protein 2 (green) from *Smn²ᴮ⁺*, *Smn²ᴮ⁻*, *+/+;Smn²ᴮ/ᶠ⁷*, and *Albᶜʳᵉ/⁺;Smn²ᴮ/ᶠ⁷* mice at P19. *Depicts occupied endplate, whereas **depicts normal neurofilament distribution; arrowheads show NF accumulation and arrows show unoccupied endplate/denervation. (**B, C**) Bar graphs show quantification of neurofilament accumulation (**B**) and endplate occupancy (**C**). n = 3, mean ± SEM. Statistical significance indicated by **$p < 0.01$, ***$p < 0.001$, following Brown-Forsythe and Welch ANOVA. Scale bar = 50 μm.

The online version of this article includes the following source data for figure 5:

**Source data 1.** Raw data for plots in *Figure 5B and C*.

## Liver-specific SMN depletion does not affect major adipocyte and gastrointestinal hormones

Gastrointestinal (GI) hormones play a key role in regulating hepatic lipid metabolism and pancreatic function (*Mells and Anania, 2013*). Thus, to assess whether adipocytes and GI hormones contribute to liver and pancreatic dysfunction in liver-specific SMN-depleted mice, we measured plasma levels of major adipocyte hormones (leptin and resistin) and GI hormones (ghrelin, gastric inhibitory polypeptide (GIP), peptide YY (PYY) and secretin) at P19 and P60 (*Mells and Anania, 2013*; *Chen et al., 2021*). Our data revealed no significant changes in adipocyte hormones at either time point (*Figure 8A–B and G–H*) across all genotypes. Moreover, few alterations in GI hormones (*Figure 8C–F, I–L*) were observed exclusively in *Smn²ᴮ⁻* mice at P19, including a marked reduction of ghrelin (*Figure 8C*) – a hormone involved in glucose homeostasis in pancreatic islet and liver (*Dong and Brubaker, 2012*) – and secretin (*Figure 8F*), which regulates secretory activity in organs such as the liver and pancreas (*Afroze et al., 2013*). Notably, no such changes were observed in *Albᶜʳᵉ/⁺;Smn²ᴮ/ᶠ* at any time point. Altogether, these results suggest that the pancreatic and liver changes at P19 are likely independent of major adipocyte and GI hormone alterations, pointing to a more direct disruption of the liver-pancreas axis in liver-specific SMN-depleted mice.

## Liver-specific SMN-depleted mice have normal survival rates and motor function

We also conducted a comprehensive assessment of survival, weight, and motor function in *Albᶜʳᵉ/⁺;Smn²ᴮ/ᶠ⁷* and *+/+;Smn²ᴮ/ᶠ⁷* mice up to postnatal day 60 (*Figure 9A*). While the *Smn²ᴮ⁻* mice

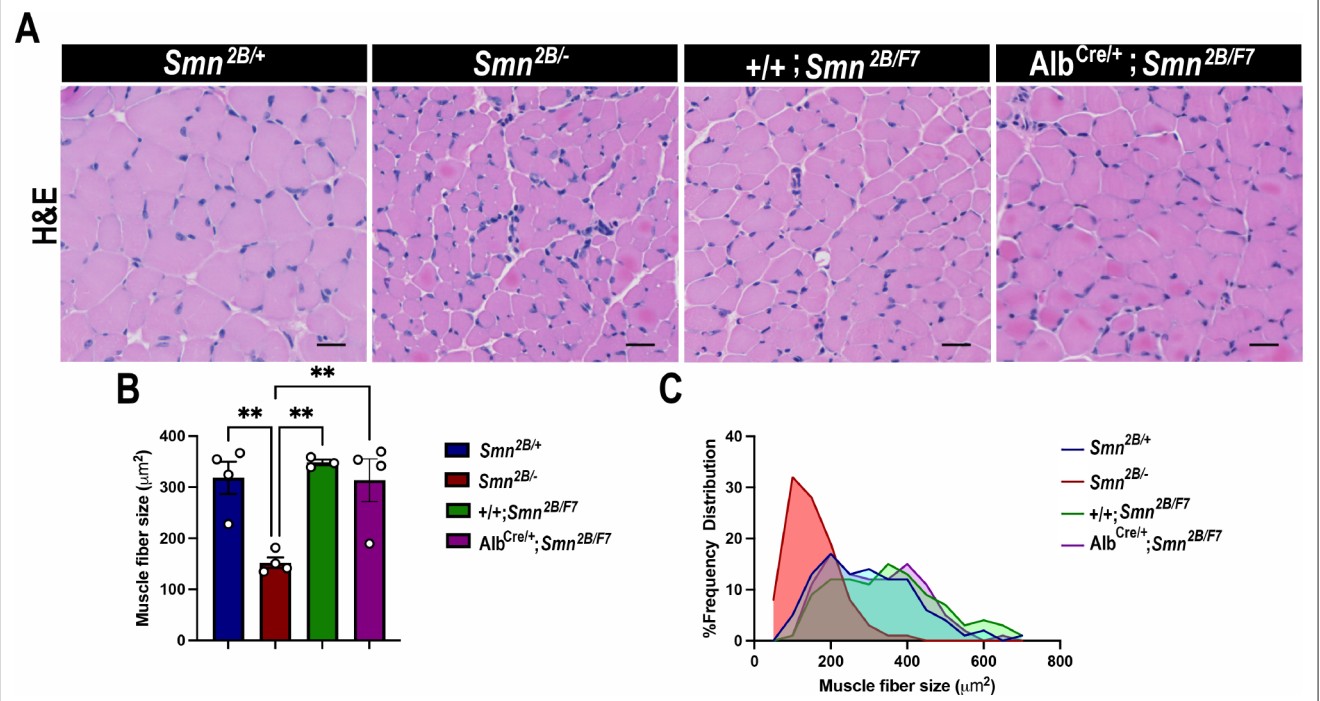

**Figure 6.** Impact of liver-specific SMN depletion on muscle fiber size and distribution. (**A**) Representative images of H&E-stained *tibialis anterior* muscle sections from *Smn*[2B/+], *Smn*[2B/-], +/+;*Smn*[2B/F7], and *Alb*[Cre/+];*Smn*[2B/F7] mice at P19. (**B**) Bar graph shows quantification of muscle fiber cross-sectional area. C. Graph demonstrating frequency of muscle fiber size across different genotypes. n ≥ 3, mean ± SEM. Statistical significance indicated by **$p < 0.01$, following Brown-Forsythe and Welch ANOVA. Scale bar = 20 μm.

The online version of this article includes the following source data for figure 6:

**Source data 1.** Raw data for plots in *Figure 6B and C*.

typically have a mean survival of 25 days (*Reilly et al., 2024*), both *Alb*[Cre/+];*Smn*[2B/F7] and +/+;*Smn*[2B/F7] mice in our study survived up to 60 days (*Figure 9B*), after which they were euthanized for tissue collection. Moreover, throughout the observation period, mice from both groups exhibited similar weight gain trends (*Figure 9C*). When weight gain at P60 was normalized to the basal weight at P7, no statistically significant differences were observed (*Figure 9D*).

To further evaluate motor function at different stages, we implemented several age-specific behavioral tests commonly used to assess motor function (*Chehade et al., 2022*; *Figure 9E–G*). We previously showed *Smn*[2B/-] mice have reduced motor function scores, as seen through increased time to right, prolonged balancing time and diminished muscle strength (*Reilly et al., 2022*). Here, we found no alterations in *Alb*[Cre/+];*Smn*[2B/F7] compared to +/+;*Smn*[2B/F7] mice in time to right, as assessed by the righting reflex (*Figure 9E*), nor in distal muscle strength measured through the inverted mesh grip test (*Figure 9F*), or motor balance and coordination evaluated by the pen test (*Figure 9G*). Overall, these findings collectively demonstrate that liver-specific SMN depletion does not impact survival, weight gain, or motor function.

## Discussion

The liver is pivotal in regulating numerous biological functions in the body. These encompass, among others, the storage and balance of lipids, carbohydrates, and iron, as well as the synthesis of vital growth factors and the clearance of non-essential or harmful metabolites (*Trefts et al., 2017*). Given its multifaceted role, any alterations in liver function can potentially have cascading effects on other organs within the body. Hence, aside from elucidating the impact of liver-specific SMN depletion on hepatic function, it is imperative to explore its effects on other tissues, as alterations in these tissues have been documented in various mouse models of SMA (*Bowerman et al., 2012b*; *Deguise et al., 2021b*; *Hua et al., 2011*; *Hua et al., 2015*).

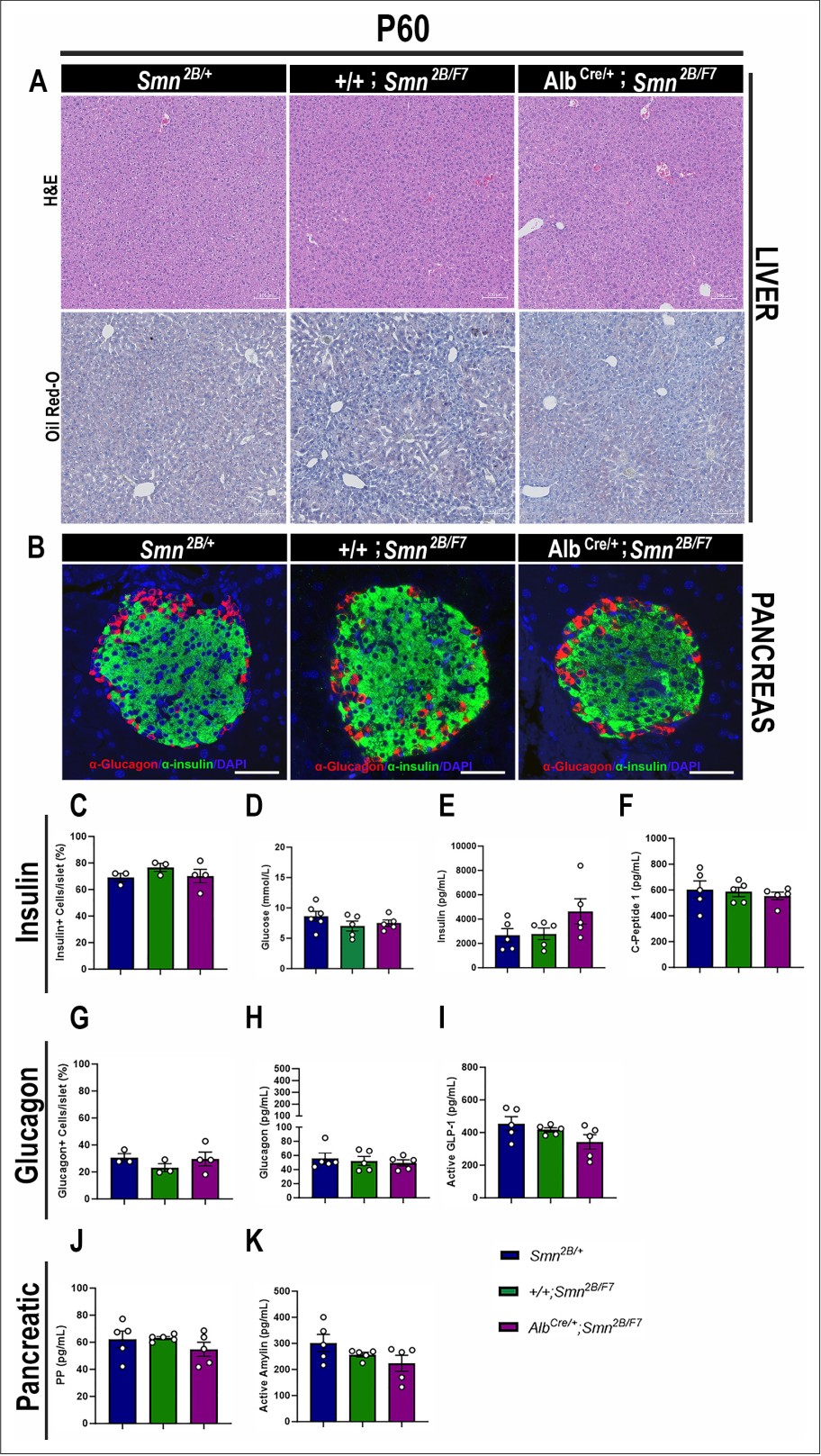

**Figure 7.** Consequences of liver-specific SMN depletion to early adulthood liver and pancreatic pathology.
(**A**) Representative images of H&E (top row) and Oil Red-O (bottom row) stained liver sections from $Smn^{2B/+}$, $+/+;Smn^{2B/F7}$, and $Alb^{Cre/+};Smn^{2B/F7}$ mice at P60. Scale bar 100 µm. (**B**) Representative immunofluorescent images of pancreatic islets stained for glucagon (red) and insulin (green) from $Smn^{2B/+}$, $+/+;Smn^{2B/F7}$, and $Alb^{Cre/+};Smn^{2B/F7}$ mice

*Figure 7 continued on next page*

*Figure 7 continued*

at P60. Scale bar 50 μm. (**C**) Bar graph shows the quantification of insulin-positive cells relative to the total number of DAPI-positive cells within the pancreatic islet. (**D**) Bar graph depicts non-fasting blood glucose levels from P60 mice across different genotypes. (**E, F**) Bar graphs illustrate non-fasting plasma levels of insulin (**E**) and C-peptide (**F**). (**G**) Bar graph shows quantification of glucagon-positive cells relative to the total number of DAPI-positive cells within the pancreatic islet. (**H–K**) Bar graphs depict non-fasting plasma levels of glucagon (**H**), active GLP-1 (**I**), pancreatic polypeptide (**J**) and active amylin (**K**). n ≥ 3, mean ± SEM. Samples were analyzed using Brown-Forsythe and Welch ANOVA, and no statistically significant differences were observed.

The online version of this article includes the following source data for figure 7:

**Source data 1.** Raw data for plots in *Figure 7C–K*.

Previous work on conditional knockout of *Smn1* in the livers of mice resulted in severe impairment of liver development characterized by iron overload and subsequent liver atrophy, ultimately leading to late embryonic lethality (*Vitte et al., 2004*). In our study, we generated a novel mouse model where instead of a complete knockout of *Smn1* in the liver, we created a scenario where SMN protein was depleted to 30% of full-length SMN production specifically within the liver, while SMN protein levels in other tissues remain comparable to those of heterozygous controls. Notably, this mouse line did not exhibit any signs of developmental delays, liver atrophy, or reduced lifespan. Nonetheless, within 19 days after birth, *Alb^Cre/+^;Smn^2B/F7^* mice showed a mild yet rapid onset of fatty liver disease and trends towards an increase in lipid levels when maintained on a normal chow diet, underscoring the importance of SMN in lipid metabolism. Interestingly, these changes were transient, as by P60, liver-specific SMN-depleted mice displayed no lipid accumulation or hepatocyte morphological changes, likely due to the compensatory proliferation of non-recombined cells.

A recent study found that 75% of pediatric and adult SMA patients showed ultrasonic evidence of mild to moderate hepatic steatosis, independent of disease severity or SMA subtype (*Leow et al., 2024*). In an in vitro model using SMA patient-derived induced pluripotent stem cell-derived hepatocyte-like cells (iHeps), a 10-fold increase in lipid accumulation was observed, correlating with reduced SMN protein expression. Proteomic and transcriptomic analyses revealed dysregulation in mitochondrial and lipid metabolism pathways, including downregulation of key genes involved in oxidative phosphorylation and fatty acid oxidation, alongside increased lipid biosynthesis. Notably, repletion of SMN in SMA iHeps restored metabolic and hepatic functions, rescuing steatosis and mitochondrial defects (*Leow et al., 2024*). Excitingly, our current work further supports these findings, demonstrating that treatment with an adeno-associated viral vector carrying a liver-specific albumin promoter in *Smn^2B/-^* mice resulted in partial restoration of liver SMN levels, improving survival, alleviating liver pathology, and restoring muscle size and pancreatic cell balance (*Sutton et al., 2024*). Additionally, a recent study found that 60.8% of adult SMA type 3 patients (23 patients) exhibited at least one lipid abnormality (*Miletić et al., 2024*), further supporting the hypothesis that hepatocyte-specific SMN deficiency contributes to liver pathology.

Mild liver steatosis may be accompanied by disruptions in crucial liver function markers, notably those implicated in iron metabolism, autophagy, and growth factor regulation, such as IGF-1 (*Deguise et al., 2021a*). Accordingly, we initially investigated the expression of HO-1, an enzyme responsible for heme degradation into iron, carbon monoxide, and biliverdin; and transferrin, a liver-synthesized glycoprotein involved in iron transport across tissues (reviewed in *Muckenthaler et al., 2017*). Prior studies have documented alterations in these proteins in *Smn^2B/-^* mouse livers (*Deguise et al., 2021b*). While our current analysis did not reveal significant differences in *Alb^Cre/+^;Smn^2B/F7^* mice, the observed upward trend in transferrin and HO levels suggests ongoing changes in iron metabolism, which may not be fully manifested at P19. In addition, incomplete Cre-mediated excision of *Smn^F7^* allele may be hindering a comprehensive understanding of SMN's role in liver function.

Furthermore, no alterations were observed for P62 levels, a multifunctional scaffold protein implicated in autophagy, formation of hepatic inclusion bodies, and hepatocyte cell death (reviewed in *Manley et al., 2013*) despite its increased presence in mouse models of SMA (*Deguise et al., 2021b*). Similarly, there were no changes detected in IGF-1 protein levels, whose deficiency has been linked to liver pathology and non-alcoholic fatty liver disease (NAFLD; *Hribal et al., 2013*; *Ichikawa et al., 2007*), and its decrease is evident in *Smn^2B/-^* mice (*Deguise et al., 2021b*). NAFLD involves the accumulation of fatty acids in more than 5% of liver cells, occurring in the absence of alcohol

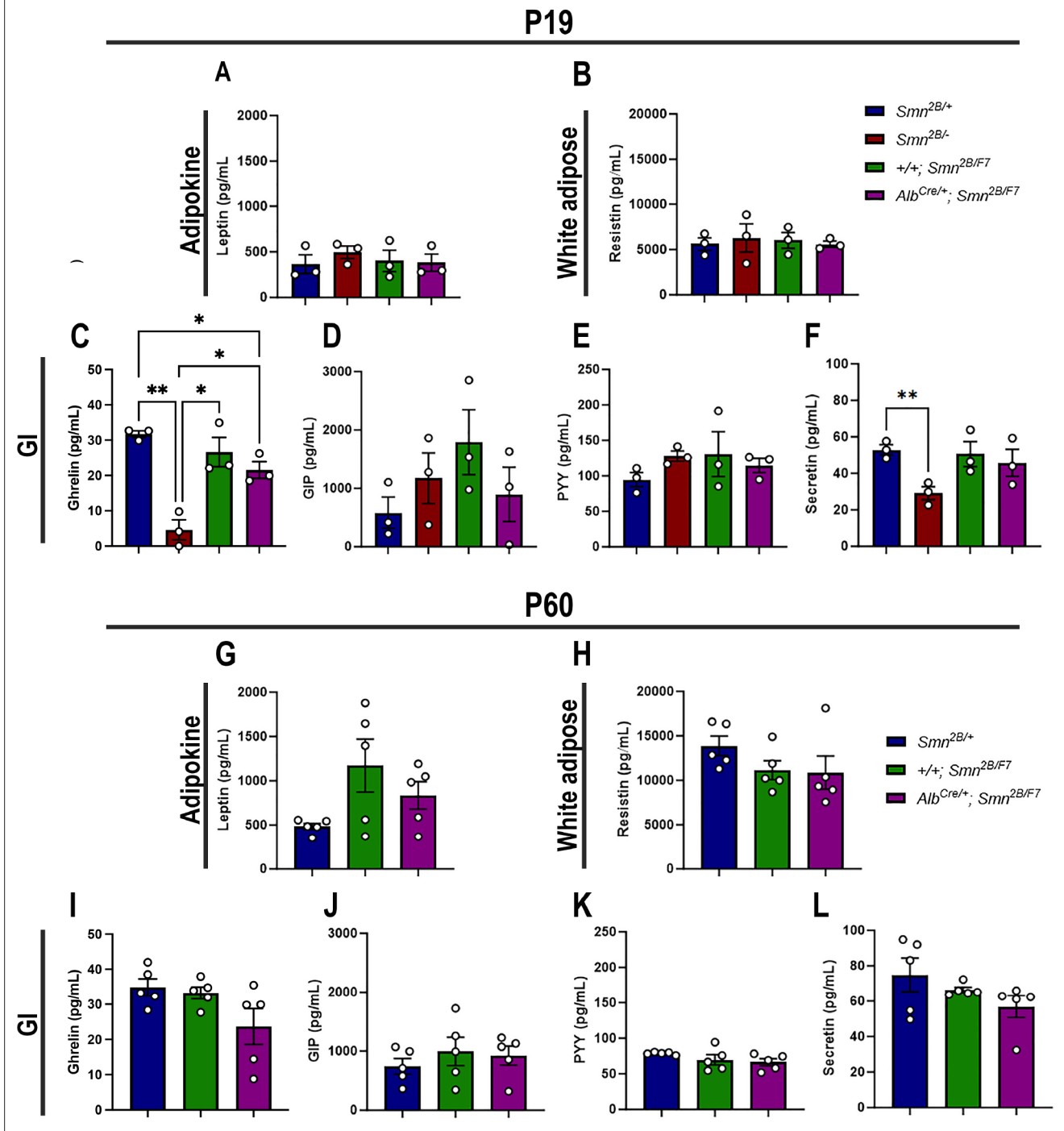

**Figure 8.** Contribution of liver-specific SMN depletion to major adipocyte and gastrointestinal hormones at P19 and P60. Panels A-B and G-H show bar graphs depicting non-fasting plasma levels of adipocyte-secreted hormones leptin (**A, G**) and resistin (**B, H**) in P19 (**A–B**) and P60 (**G–H**) mice across different genotypes. Panels C-F and I-L illustrate non-fasting plasma levels of gastrointestinal-secreted hormones, including ghrelin (**C, I**), gastric inhibitory polypeptide (**D, J**), peptide YY (**E, K**), and secretin (**F, L**) in P19 (**C–F**) and P60 (**I–L**) mice. n ≥ 3, mean ± SEM. Statistical significance is indicated by *p < 0.05, **p < 0.01, following Brown-Forsythe and Welch ANOVA.

The online version of this article includes the following source data for figure 8:

**Source data 1.** Raw data for plots in *Figure 8*.

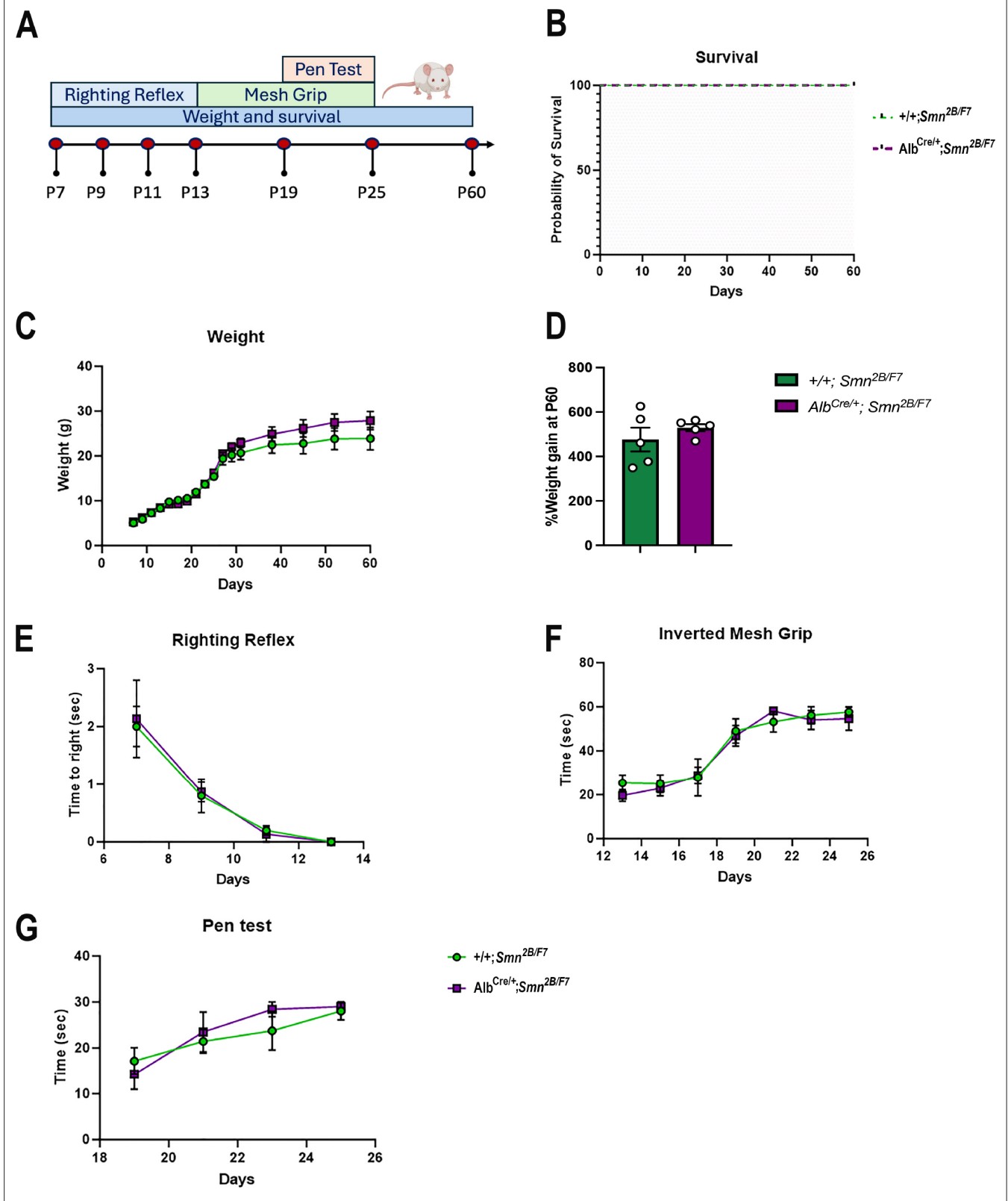

**Figure 9.** Impact of liver-specific SMN depletion on motor function. (**A**) Schematic representation of experimental design. The righting reflex test was conducted from P7 to P13, the inverted mesh grip test from P13 to P25, and the pen test from P19 to P25. Assessments were performed every 2 days. Weight was measured every 2 days until day 30, and then weekly until day 60. Animal welfare was monitored throughout the 60-day period during weight measurements. (**B**) Kaplan–Meier survival curve comparing $Alb^{Cre/+};Smn^{2B/F7}$ and $+/+;Smn^{2B/F7}$ mice up to 60 days. (**C, E-G**) Graphs show weight

*Figure 9 continued on next page*

*Figure 9 continued*

(**C**), righting reflex (**E**), inverted mesh grip (**F**) and pen test (**G**). n=5 per genotype, mean ± SEM. (**B**) Kaplan–Meier survival analysis; (**C, E-G**) two-way ANOVA, followed by Šídák's method. (**D**) Percentage weight gain at P60, normalized to basal weight at P7. n=5 per genotype, mean ± SEM, two-tailed unpaired Student's t-test.

The online version of this article includes the following source data for figure 9:

**Source data 1.** Raw data for plots in *Figure 9B–G*.

consumption. This condition presents with varying degrees of severity, with simple steatosis being the most common manifestation. Simple steatosis is characterized by the accumulation of fat in the liver with minimal impact on liver function (*Loomba and Sanyal, 2013*; *Tandra et al., 2011*). Consequently, our model holds promise as a valuable resource for investigating microvesicular steatohepatitis and NAFLD during the early postnatal period. Additionally, unlike the well-studied $Smn^{2B/-}$ model of SMA, this liver-specific SMN depletion enables the study of hepatic SMN function without the confounding effects of systemic SMN depletion or the severe phenotype observed in the $Smn^{2B/-}$ mice.

Previously, our research unveiled metabolic abnormalities in SMA-like mice, including glucose intolerance, insulin hypersensitivity, and hyperglucagonemia, accompanied by changes in pancreatic islet composition characterized by an increase in glucagon-producing α-cells at the expense of insulin-producing β-cells (*Bowerman et al., 2012b*). Histopathological examination of pancreatic tissue from infants and children with SMA type I corroborated these findings, revealing a predominant presence of glucagon-producing α-cells within pancreatic islets (*Bowerman et al., 2012b*). Furthermore, one-year-old $Smn1^{+/-}$ +/- producing approximately 50% of SMN protein, when exposed to a high-fat diet, exhibited dysregulation in the proportion of glucagon-producing α-cells within pancreatic islets and heightened hepatic insulin and glucagon sensitivity (*Bowerman et al., 2014*).

Here, we uncovered a novel intrinsic role of liver-specific SMN in pancreatic function, distinct from IGF-1 signaling, pancreatic SMN levels (which remained stable), or canonical SMA pathology. Although the precise role of SMN in the liver-pancreas axis remains unclear, we speculate that altered amino acid metabolism may contribute to the disrupted communication between these two organs. The liver plays a central role in amino acid metabolism, and disturbances in this process can affect the liver-α cell axis, a well-documented feedback loop in which circulating amino acids stimulate glucagon secretion, which in turn regulates hepatic amino acid uptake and metabolism (*Richter et al., 2022*). In the context of metabolic diseases such as NAFLD and SMA, liver dysfunction, including impaired lipid metabolism, may disrupt this axis, leading to abnormal glucagon secretion (*Richter et al., 2022*; *Deng et al., 2024*). It is possible that dysregulated amino acid pathways may directly influence this feedback loop.

Further evidence for this hypothesis comes from a recent study in individuals with type 2 diabetes (T2D) and NAFLD, which found hyperglucagonaemia and altered amino acid profiles. In T2D-NAFLD patients, plasma glucagon levels were significantly higher than in individuals without NAFLD, with a positive correlation between plasma glucagon concentrations and liver fat content. This study also showed increased plasma levels of isoleucine and glutamate, and decreased levels of glycine and serine in individuals with NAFLD (*Rix et al., 2024*). These findings suggest that metabolic dysregulation in the liver, including alterations in amino acid metabolism, may exacerbate glucagon secretion. Interestingly, such changes in amino acid metabolism are consistent with clinical observations in SMA patients, where an 'amino acid diet' — low in fats and rich in specific amino acids — has been shown to provide some clinical benefit (*O'Connor et al., 2023*). Given this, it is plausible that impaired liver metabolism in SMN-depleted mice alters the liver-α cell axis, leading to dysregulated glucagon and GLP-1 secretion. While this proposed link between liver steatosis, amino acid metabolism, and pancreatic dysfunction is speculative, we are actively investigating the mechanisms underlying this liver-pancreas interplay in mouse models of SMA.

In addition, while the hallmark pathological features of SMA include motor neuron loss, NMJ pathology, muscle atrophy, and consequent motor function alterations, these were not evident in the $Alb^{Cre/+};Smn^{2B/F7}$ mouse model at P19. However, given that glucose metabolism defects are observed in various neurodegenerative and neuromuscular disorders, subsequent work is required to clarify the impact of liver-specific SMN depletion on pancreatic function. Although hepatic and pancreatic abnormalities likely occur independently of disease onset, given that SMN depletion in motor neurons and muscle alone is sufficient to induce an SMA-like phenotype (*Frugier et al., 2000*; *Cifuentes-Diaz*

**Table 1.** List of primers used for genotyping.

| Name | Sequence (5'–3') |
| --- | --- |
| | TGCAAACATCACATGCACAC |
| | TTGGCCCCTTACCATAACTG |
| Alb-Cre | GAAGCAGAAGCTTAGGAAGATGG |
| | AGAAGGAAAGTGCTCACATACAAATT |
| SmnF7 | TGTCTATAATCCTCATGCTATGGAG |
| | GGTTTCAGACAAAATAAAAAAGAATTTAAG |
| Smn2B | TTTGGCAGACTTTAGCAGGGC |

*et al., 2001*; *Cifuentes-Diaz et al., 2002*), the potential influence of these anomalies on SMA progression warrants careful consideration.

Overall, our findings suggest that alterations in SMN production solely within the liver may suffice to induce pathological hepatic and pancreatic function, shedding light on a novel role of SMN in liver physiology.

# Materials and methods

## Mouse models

The $Smn^{2B}$ mouse, which harbors the $Smn^{2B}$ mutation resulting from the substitution of three nucleotides within the exon splicing enhancer of exon 7, was developed and maintained on a C57BL/6 background in our laboratory (*Bowerman et al., 2012a*; *Eshraghi et al., 2016*). The FVB.129(B6)-$Smn1^{tm1Jme}$ ($Smn^{F7/F7}$) (RRID:IMSR_JAX:006138) and B6.Cg-$Speer6$-$ps1^{Tg(Alb-Cre)21Mgn}$/J ($Alb^{Cre}$) (RRID:IMSR_JAX:003574) mice were acquired from The Jackson Laboratory. Liver-specific SMN-depleted mice were obtained through successive crosses between $Alb^{Cre/+}$;$Smn^{F7/+}$ and $Smn^{2B/F7}$ mice (see *Figure 1*). Line validation was conducted via genotyping of DNA extracted from mouse ear biopsies using PCR (see *Table 1* for list of primers used). Both male and female mice were included in the studies. All mice were bred and housed in the University of Ottawa Animal Facility under protocol OHRI-3343, adhering to guidelines established by the Canadian Council on Animal Care. Additionally, we employed $Smn^{2B/-}$ mice (a severe SMA mouse model) and asymptomatic heterozygous $Smn^{2B/+}$ mice as controls in our experiments.

## Tissue harvesting and processing

Following euthanasia, liver, and *tibialis anterior* (TA) muscles underwent fixation in a 1:10 dilution of buffered formalin for 48 hr at 4 °C, followed by transfer to 70% ethanol at the same temperature until processing. The pancreas was fixed in 4% paraformaldehyde (PFA) for 48 hr at 4 °C, then similarly moved to 70% ethanol for processing. For the lumbar spinal cord (SC) and liver (for Oil Red-O staining), fixation was performed overnight at 4 °C in 4% PFA, followed by immersion in 30% sucrose in PBS. Subsequently, the liver and spinal cord were flash-frozen in optimal cutting temperature (OCT) embedding medium. Cryosections of the lumbar spinal cord were cut at a thickness of 16 μm, while liver sections were cut at 10 μm and stored at –80 °C until immunohistochemical analysis. Cre immunostaining of cryosectioned liver and spinal cord was performed as described in section 4.4.1, with the exception that the antigen retrieval step was not performed.

The abdominal musculature was dissected and fixed in 4% PFA for 15 min at room temperature, followed by 3 washes in 1 X PBS, after which the transversus abdominis (TVA) muscle was dissected from the abdominal musculature. TA, liver, and pancreas samples were handled at the University of Ottawa Department of Pathology and Laboratory Medicine. Paraffin blocks containing tissues were sectioned at 4 μm thickness using a microtome. Sections of TA and liver were stained with hematoxylin and eosin (H&E) using an XL CV5030 autostainer from Leica. Liver cryosections were stained with Oil Red-O and counterstained with hematoxylin using established protocols. Images of H&E and Oil Red-O-stained samples were scanned with a MIRAX MIDI digital slide scanner manufactured by Zeiss. Image acquisition was performed using Panoramic Viewer 1.15.4 software.

## Immunoblotting

Total protein was extracted by homogenizing flash-frozen samples of spinal cord, brain, liver, pancreas and hindlimb muscles using RIPA lysis buffer (Cell Signaling), supplemented with phenylmethylsulfonyl fluoride (PMSF, Cell Signaling). The concentrations of the extracted proteins were determined using the Pierce BCA Protein Assay kit (ThermoFisher).

For liver, brain, and spinal cord tissues, 20 µg of protein per sample were separated by 10% sodium dodecyl sulfate polyacrylamide gel electrophoresis (SDS-PAGE), while 30 µg were used for pancreas and muscle samples. Following electrophoresis, the proteins were transferred onto a PVDF membrane (Immobilon-P, Millipore) and blocked for 30 min at room temperature using intercept blocking buffer (LI-COR). Subsequently, the membranes were incubated overnight at 4 °C with primary antibodies.

After incubation, the membranes were rinsed with tris-buffered saline containing 0.1% Tween-20 (TBS-T) and then incubated for 1 hr at room temperature with IRDye 680 R or 800CW (LI-COR) secondary antibody. Fluorescent signals were detected using the Odyssey Infrared Imaging System (LI-COR). Total raw values were normalized to α-tubulin (housekeeping protein) obtained from the same blot.

## Immunohistochemistry

### Pancreas

Pancreatic sections underwent deparaffinization using three washes of Histo-Clear (National Diagnostics, Atlanta, GA) for 5 min each, followed by two washes of a 50/50 mixture of absolute ethanol and Histo-Clear for 3 min each. Subsequently, the slides were gradually rehydrated in two washes of 100% (v/v) ethanol (3 min each), followed by one wash of 95% (v/v), 70% (v/v), 50% (v/v), and 0% (v/v) ethanol

**Table 2.** List of antibodies used.

| Application | Antibody | Species | Dilution | Company (Catalog #) |
|---|---|---|---|---|
| Western Blot | SMN | Mouse | 1:2 500 | BD Transduction (610647) |
| Western Blot | Alpha-tubulin | Rabbit | 1:10 000 | Abcam (ab4074) |
| Western Blot | Alpha-tubulin | Mouse | 1:10 000 | Sigma-Aldrich (CP06) |
| Western Blot | Heme oxygenase-1 (HO) | Rabbit | 1:10 000 | Abcam (ab68477) |
| Western Blot | P62 | Mouse | 1:1 000 | Abcam (ab56416) |
| Western Blot | Transferrin | Rabbit | 1:1 000 | Abcam (ab82411) |
| Western Blot | Goat anti-mouse IR Dye 800 CW | | 1:5 000 | LI-COR (925–32210) |
| Western Blot | Goat anti-mouse IR Dye 680 R | | 1:5 000 | LI-COR (926–68070) |
| Western Blot | Goat anti-rabbit IR Dye 800 CW | | 1:5 000 | LI-COR (925–32211) |
| Western Blot | Goat anti-rabbit IR Dye 680 R | | 1:5 000 | LI-COR (926–68071) |
| IHC | ChAT | Goat | 1:100 | Millipore (AB144P) |
| IHC | Donkey anti-goat Alexa Fluor 555 | | 1:200 | Invitrogen (A21432) |
| IHC | TRITC conjugated bungarotoxin | N/A | 1:1 000 | Invitrogen (T1175) |
| IHC | Neurofilament (NF-M) | Mouse | 1:100 | (Developmental Studies Hybridoma Bank, P12839) |
| IHC | Synaptic vesicle glycoprotein 2 A (SV2A) | Mouse | 1:250 | (Developmental Studies Hybridoma Bank, Q7L0J3) |
| IHC | Goat anti-mouse Alexa Fluor 488 | | 1:250 | Invitrogen (A11001) |
| IHC | Glucagon | Mouse | 1:200 | Abcam (ab10988) |
| IHC (Parafin) IHC (Frozen) | Cre Recombinase (D7L7L) | Rabbit | 1:500 1:200 | Cell Signaling (15036) |
| IHC | Insulin | Rabbit | 1:50 | Abcam (ab181547) |
| IHC | Goat anti-mouse Alexa Fluor 555 | | 1:500 | Invitrogen (A21422) |
| IHC | Goat anti-rabbit Alexa Fluor 488 | | 1:500 | Invitrogen (A11034) |

for 3 min each. The slides were then incubated in 0.5% Triton-X-100 (Millipore Sigma, Burlington, MA) in PBS for 5 min, followed by three washes with PBS. Afterward, the slides were blocked in a solution containing 20% goat serum and 0.3% Triton-X-100 in TBS for 2 hr.

Primary antibodies targeting insulin, glucagon or Cre (*Table 2*) were applied to the slides in a solution containing 2% goat serum and 0.3% Triton-X-100 in TBS overnight at 4 °C. The slides were then washed three times with PBS and subsequently incubated with secondary antibodies in a solution containing 2% goat serum and 0.3% Triton-X-100 for 1 hr. Following this, the solution was removed, and DAPI (1:1000) in PBS was added for 5 min at room temperature. The slides were washed three times with PBS and mounted in Fluoromount-G Mounting Medium (Invitrogen).

Images were taken using an Axio Imager M2 microscope (Zeiss) and captured with a 40 X objective. The number of glucagon and insulin-positive cells per islet was counted for each mouse. Analysis was blinded where possible.

## Motor neurons

Lumbar spinal cord sections were prepared for choline acetyltransferase (ChAT) staining of motor neurons. The slides were air-dried at room temperature for 30 min and then rinsed in TBS-T for 5 min. Samples were permeabilized in 0.3% Triton X-100 in PBS for 30 min, followed by blocking in 1 x Power Block (BioGenex, Fremont, CA) for 20 min at room temperature. Subsequently, the samples were incubated with a goat anti-ChAT (details in *Table 2*) antibody at a dilution of 1:100 in 1% BSA and 0.3% Triton X-100 in PBS for 3 nights at 4 °C.

After the initial antibody incubation, slides were washed twice for 10 min with TBS-T at room temperature. Following the washes, samples were then incubated with Alexa Fluor 555 donkey anti-goat IgG at a dilution of 1:200 in 10% donkey serum and 0.3% Triton X-100 in PBS for 2 hr at room temperature in a humid chamber. Nuclei were counterstained with 4',6-diamidino-2-phenylindole (DAPI) at a dilution of 1:1000 in PBS for 5 min.

Finally, the slides were rinsed 3 X for 10 min with TBS-T and mounted with Fluoromount-G Mounting Medium (Invitrogen). Spinal cord sections were examined under fluorescence using an Axio Imager M2 microscope (Zeiss) and imaged at ×20 magnification for quantification, and at ×63 for representative pictures. The number of ChAT-positive motor neurons with a diameter of ≥20 μm per ventral horn was recorded for 7–10 different sections per animal, each separated by at least 100 μm to prevent re-counting of the same motor neuron. An average number of motor neuron cell bodies per section was determined. Analysis was blinded where possible.

## Neuromuscular junction (NMJ)

To evaluate neuromuscular innervation, we performed immunohistochemistry using established protocols (*Reilly et al., 2022*). After euthanasia, the TVA muscles were promptly dissected and fixed in 4% paraformaldehyde (Electron Microscopy Science) in PBS for 15 min. Post-synaptic acetylcholine receptors (AChRs) were labeled for 10 min with alpha-bungarotoxin (aBTX) conjugated to Alexa Fluor 488. Subsequently, the muscles were blocked in 4% bovine serum albumin (BSA) and 1% Triton-X-100 in PBS for 30 min.

Following blocking, the muscles underwent overnight incubation with primary antibodies for neurofilament and synaptic vesicle protein 2 (refer to *Table 2*). Visualization of the labeled structures was achieved using DyLight-conjugated secondary antibodies. The muscles were then whole-mounted in Dako Fluorescent mounting media and imaged with a 20 x objective on a Zeiss Axio Imager M1 microscope.

A minimum of three fields of view (FOV) were quantified per muscle. For each FOV, the percentage of fully occupied endplates was noted. Fully occupied endplates were defined as motor endplates completely covered by the presynaptic terminal labeled with synaptic vesicle protein 2 (SV2) and neurofilament (NF). Analysis was blinded where possible.

## Muscle fiber analysis

Muscle fiber analysis was performed using ImageJ software (version 2.9.0/1.53t). A total of one hundred fibers were evaluated per animal, covering various regions of the muscle section to ensure good representation. The area of each fiber was measured in square micrometers (μm²) to compute both the average and distribution of fiber sizes for each animal.

## Glucose levels

Non-fasting blood glucose concentrations were assessed using a OneTouch Ultra2 Blood Glucose Test Strips (LifeScan Europe GmbH, Zug, Switzerland) immediately following blood collection. Approximately 2 µL of blood was applied to the test strip for glucose concentration measurement.

## ELISA for insulin-like growth factor I (IGF-1)

The concentration of IGF-1 in the liver was quantified using the Mouse/Rat IGF-I/IGF1 Quantikine ELISA Kit (MG100, R&D Systems, Inc, Minneapolis, MN, USA). Liver protein lysates were diluted at a ratio of 1:6 in calibrator diluent and the assay was performed following the manufacturer's instructions.

## Triglycerides quantification

Liver tissues from P19 mice were promptly dissected and flash frozen. Subsequently, triglyceride analysis was conducted at the Vanderbilt Mouse Metabolic Phenotyping Center, employing established protocols previously utilized by our team (*Deguise et al., 2021b*).

## Mouse metabolic hormone assay

Blood was collected from the mice immediately after decapitation using a Microcuvette CB 300 K2E coated with K2 EDTA (16.444.100) for capillary sampling. All samples were taken randomly without a fasting period. The blood was centrifuged at $5000 \times g$ for 5 min at room temperature to separate the plasma. For plasma volumes less than 50 µL, genotypes were pooled to achieve a minimum volume of 50 µL as needed. The multiplexing analysis was performed using the Luminex 200 system (Luminex, Austin, TX, USA) by Eve Technologies Corp. (Calgary, Alberta). Twelve markers were simultaneously measured in the samples using Eve Technologies' Mouse Metabolic Hormone 12-Plex Discovery Assay (MilliporeSigma, Burlington, Massachusetts, USA) according to the manufacturer's protocol. The 12-plex consisted of Amylin(active), C-Peptide 2, Ghrelin, GIP(total), GLP-1(active), Glucagon, Insulin, Leptin, PP, PYY, Resistin and Secretin. Assay sensitivities of these markers range from 1.4 to 91.8 pg/mL for the 12-plex. Individual analyte sensitivity values are available in the MilliporeSigma MILLIPLEX MAP protocol. If analyte levels were too low to be detected and fell outside the dynamic range, they were recorded as 0 in the graphs.

## Motor function, weight, and survival assessments

Motor function, weight, and survival were assessed according to established protocols (*Chehade et al., 2022*). Spinal reflexes were evaluated using the righting reflex test, while motor balance and coordination were evaluated with the pen test. Additionally, muscle strength was measured using the inverted mesh grip test (see *Figure 9A*). The righting reflex test was conducted from P7 to P13, the inverted mesh grip test from P13 to P25, and the pen test from P19 to P25. Assessments were performed every 2 days. A maximum duration of 30 s for the pen test and 60 s for the inverted mesh grip represented the established threshold. Three consecutive measurements were obtained, and their average was recorded for subsequent analysis. Weight was measured every 2 days starting at day 7 until day 30, and then weekly until day 60. Animal welfare was monitored throughout the 60-day period during weight measurements.

## Statistical analysis

Survival data were visualized using Kaplan-Meier survival curves, and intergroup differences were evaluated using the Mantel-Cox test. Brown-Forsythe and Welch ANOVA were employed to compare three or more sets of unpaired measurements, given the mosaic-like outcome observed in the $Alb^{Cre/+};Smn^{2B/F7}$ model, where assumptions of equal variances were not tenable. Two-way ANOVA, followed by Šídák's method for multiple comparisons, was utilized to assess the influence of two factors on a response. Pairwise comparisons were performed using a two-tailed Student's t-test. Statistical analyses were conducted using GraphPad Prism V.10.2.1 (GraphPad Prism Software, San Diego, CA), with significance set at $p < 0.05$. Data were presented as mean ± standard error of the mean. Sample sizes (n) are provided in figure legends, indicating the number of biological replicates analyzed, with each data point corresponding to an individual mouse from at least two different litters. Detailed statistical information is outlined in the respective figure legends, where statistical significance is indicated by *, $p < 0.05$; **, $p < 0.01$; ***, $p < 0.001$.

## Data sharing

All authors had access to the study data and reviewed and approved the final manuscript. All data associated with this study are available in the main text or supplementary materials. Imaging data used in this manuscript have been deposited at the BioImage Archive repository under Accession S-BIAD1633.

## Acknowledgements

This work was supported by Muscular Dystrophy Association (USA) [grant number 963652 to RK]; the Canadian Institutes of Health Research [grant number PJT-186300 to RK]; and the University of Ottawa Brain and Mind Institute TRIMS Award to MMAA.

## Additional information

### Funding

| Funder | Grant reference number | Author |
| --- | --- | --- |
| Muscular Dystrophy Association | 10.55762/pc.gr.157042 | Rashmi Kothary |
| Canadian Institutes of Health Research | PJT-186300 | Rashmi Kothary |
| University of Ottawa Brain and Mind Institute | TRIMS Award | Monique Marylin Alves de Almeida |

The funders had no role in study design, data collection and interpretation, or the decision to submit the work for publication.

### Author contributions

Monique Marylin Alves de Almeida, Conceptualization, Data curation, Formal analysis, Investigation, Methodology, Writing – original draft, Writing – review and editing; Yves De Repentigny, Sabrina Gagnon, Investigation, Methodology; Emma R Sutton, Formal analysis, Investigation, Visualization, Methodology; Rashmi Kothary, Conceptualization, Supervision, Funding acquisition, Project administration, Writing – review and editing

### Author ORCIDs

Monique Marylin Alves de Almeida ⓘ https://orcid.org/0000-0002-8594-1410
Rashmi Kothary ⓘ https://orcid.org/0000-0002-9239-7310

### Ethics

All mice were bred and housed in the University of Ottawa Animal Facility under protocol OHRI-3343, adhering to guidelines established by the Canadian Council on Animal Care.

Reviewer #1 (Public review): https://doi.org/10.7554/eLife.99141.3.sa1
Reviewer #2 (Public review): https://doi.org/10.7554/eLife.99141.3.sa2
Author response https://doi.org/10.7554/eLife.99141.3.sa3

## Additional files

### Supplementary files

MDAR checklist

### Data availability

All data generated or analysed during this study are included in the manuscript and supporting files; source data files have been provided for *Figures 1 and 2* (both are for the immunoblots). Note that this study did not have any numerical source data. *Figure 1—source data 1* and *Figure 2—source*

*data 1* contain the original blots data used to generate the western blot images in the figures. In addition, source data for *Figures 1–9* in the form of Excel files have been provided. These contain the raw data used to generate the plots/graphs in the manuscript. All images used to generate the figures (*Figures 2–7*) have been deposited at the BioImage Archive repository under accession S-BIAD1633.

The following dataset was generated:

| Author(s) | Year | Dataset title | Dataset URL | Database and Identifier |
|-----------|------|---------------|-------------|-------------------------|
| Kothary R | 2025 | Image data accompanying the paper "Impact of liver-specific survival motor neuron (SMN) depletion on central nervous system and peripheral tissue pathology" | https://doi.org/10.6019/S-BIAD1633 | BioImage Archive, 10.6019/S-BIAD1633 |

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
