## [Editor Report · eLife Assessment]

This work presents an **important** mouse model for a liver-specific depletion of the Survival Motor Neuron (SMN) protein, where the liver retains 30% of functional full-length SMN protein. The authors provide a profile of phenotypic changes in liver-specific SMN depleted mice with **convincing** evidence supporting their claims.

---

## [Referee Report · Reviewer #1 (Public review)]

Summary:

This manuscript presents a comprehensive exploration of the role of liver-specific Survival Motor Neuron (SMN) depletion in peripheral and central nervous system tissue pathology through a well-constructed mouse model. This study is pioneering in its approach, focusing on the broader physiological implications of SMN, which has traditionally been associated predominantly with spinal muscular atrophy (SMA).

Strengths:

(1) Novelty and Relevance: The study addresses a significant gap in understanding the role of liver-specific SMN depletion in the context of SMA. This is a novel approach that adds valuable insights into the multi-organ impact of SMN deficiency.

(2) Comprehensive Methodology: The use of a well-characterized mouse model with liver-specific SMN depletion is a strength. The study employs a robust set of techniques, including genetic engineering, histological analysis, and various biochemical assays.

(3) Detailed Analysis: The manuscript provides a thorough analysis of liver pathology and its potential systemic effects, particularly on the pancreas and glucose metabolism.

(4) Clear Presentation: The manuscript is well written. The results are presented clearly with well-designed figures and detailed legends.

Weaknesses:

(1) Limited Time Points: The study primarily focuses on a single time point (P19). This limits the understanding of the temporal progression of liver and pancreatic pathology in the context of SMN depletion. Longitudinal studies would provide a better understanding of disease progression.

(2) Incomplete Recombination: The mosaic pattern of Cre-mediated excision leads to variability in SMN depletion, which complicates the interpretation of some results. Ensuring more consistent recombination across samples would strengthen the conclusions.

After the revision, the authors addressed the reviewers' questions by extending their analyses to include P60 mice, conducting both liver and pancreatic analyses, and adding a comprehensive panel of metabolic hormones related to glucose metabolism in animals at P19 and P60. They also corrected all errors identified during the initial review process and expanded the discussion to clarify raised issues. All my questions have now been addressed.

---

## [Referee Report · Reviewer #2 (Public review)]

Summary:

Marylin Alves de Almeida et al. developed a novel mouse cross via conditionally depleting functional SMN protein in the liver (AlbCre/+;Smn2B/F7). This mouse model retains a proportion of SMN in the liver, which better recapitulates SMN deficiency observed in SMA patients and allows further investigation into liver-specific SMN deficiency and its systemic impact. They show that AlbCre/+;Smn2B/F7 mice do not develop an apparent SMA phenotype as mice did not develop motor neuron death, neuromuscular pathology or muscle atrophy, which is observed in the Smn2B/- controls. Nonetheless, at P19 and P60, these mice develop mild liver steatosis, and interestingly, this conditional depletion of SMN in the liver impacts cells in the pancreas.

Strengths:

The current model has clearly delineated the apparent metabolic perturbations which involve a significantly increased lipid accumulation in the liver and pancreatic cell defects in AlbCre/+;Smn2B/F7 mice at P19 and P60. Standard methods like H&E and Oil Red-O staining show that in AlbCre/+;Smn2B/F7 mice, their livers closely mimic the livers of Smn2B/- mice, which have the full body knockout of SMN protein. Unlike previous work, this liver-specific conditional depletion of SMN is superior in that it is not lethal to the mouse, which allows an opportunity to investigate the long-term effects of liver-specific SMN on the pathology of SMA.

Weaknesses:

Given that SMA often involves fatty liver, dyslipidemia and insulin resistance, using the current mouse model, the authors could have explored the long-term effects of liver-specific depletion of SMN on metabolic phenotypes beyond P19, as well as systemic effects like glucose homeostasis. Given that the authors also report pancreatic cell defects, the long-term effect on insulin secretion and resistance could be further explored. This has been addressed in the revised manuscript. The mechanistic link between a liver-specific SMN depletion and apparent pancreatic cell defects has been made clearer.

Discussion:

This current work explores a novel mouse cross in order to specifically deplete liver SMN using an Albumin-Cre driver line. This provides insight into the contribution of liver-specific SMN protein to the pathology of SMA, which is relevant for understanding metabolic perturbations in SMA patients. Nonetheless, given that SMA in patients involve a systemic deletion or mutation of the SMN gene, the authors could emphasize the utility of this liver-specific mouse model, as opposed to using in vitro models, which have been recently reported (Leow et al, 2024, JCI).

Comments on current version:

No further suggestions. Previous recommendations have been addressed by the authors.

---

## [Author Response]

The following is the authors’ response to the original reviews.

**Public Reviews:**

**Reviewer #1 (Public review):**
Summary:This manuscript presents a comprehensive exploration of the role of liver-specific Survival Motor Neuron (SMN) depletion in peripheral and central nervous system tissue pathology through a well-constructed mouse model. This study is pioneering in its approach, focusing on the broader physiological implications of SMN, which has traditionally been associated predominantly with spinal muscular atrophy (SMA).Strengths:(1) Novelty and Relevance: The study addresses a significant gap in understanding the role of liver-specific SMN depletion in the context of SMA. This is a novel approach that adds valuable insights into the multi-organ impact of SMN deficiency.(2) Comprehensive Methodology: The use of a well-characterized mouse model with liver-specific SMN depletion is a strength. The study employs a robust set of techniques, including genetic engineering, histological analysis, and various biochemical assays.(3) Detailed Analysis: The manuscript provides a thorough analysis of liver pathology and its potential systemic effects, particularly on the pancreas and glucose metabolism.(4) Clear Presentation: The manuscript is well written. The results are presented clearly with well-designed figures and detailed legends.

We thank the reviewer for their positive comments. They had some concerns for us to consider (see below). We provide a point-by-point response to their comments.

Weaknesses:(1) Limited Time Points: The study primarily focuses on a single time point (P19). This limits the understanding of the temporal progression of liver and pancreatic pathology in the context of SMN depletion. Longitudinal studies would provide a better understanding of disease progression.

We thank the reviewer for the suggestion. We extended our analysis to include P60 mice and performed both liver and pancreatic analyses at this time point to address this suggestion.

(2) Incomplete Recombination: The mosaic pattern of Cre-mediated excision leads to variability in SMN depletion, which complicates the interpretation of some results. Ensuring more consistent recombination across samples would strengthen the conclusions.

The variability in Cre-mediated excision is inherently stochastic, influenced by factors such as Cre expression levels, timing of recombination, and the accessibility of the target locus in individual cells. Achieving complete consistency across samples is particularly challenging, especially given the complexity of our breeding scheme, which occasionally results in litters without any animals of the desired genotype. Importantly, our study not only demonstrates that liver-specific SMN depletion results in liver alterations and pancreatic dysfunction but also highlights the limitations and challenges associated with this mouse model. By doing so, we aim to provide valuable insights for other researchers considering similar approaches in future studies.

**Reviewer #2 (Public review):**
Summary:Marylin Alves de Almeida et al. developed a novel mouse cross via conditionally depleting functional SMN protein in the liver (AlbCre/+;Smn2B/F7). This mouse model retains a proportion of SMN in the liver, which better recapitulates SMN deficiency observed in SMA patients and allows further investigation into liver-specific SMN deficiency and its systemic impact. They show that AlbCre/+;Smn2B/F7 mice do not develop an apparent SMA phenotype as mice did not develop motor neuron death, neuromuscular pathology or muscle atrophy, which is observed in the Smn2B/- controls. Nonetheless, at P19, these mice develop mild liver steatosis, and interestingly, this conditional depletion of SMN in the liver impacts cells in the pancreas.Strengths:The current model has clearly delineated the apparent metabolic perturbations which involve a significantly increased lipid accumulation in the liver and pancreatic cell defects in AlbCre/+;Smn2B/F7 mice at P19. Standard methods like H&E and Oil Red-O staining show that in AlbCre/+;Smn2B/F7 mice, their livers closely mimic the livers of Smn2B/- mice, which have the full body knockout of SMN protein. Unlike previous work, this liver-specific conditional depletion of SMN is superior in that it is not lethal to the mouse, which allows an opportunity to investigate the long-term effects of liver-specific SMN on the pathology of SMA.

We thank the reviewer for their positive comments. They had some concerns for us to consider (see below). We provide a point-by-point response to their comments (review comments in black, our response in red).

Weaknesses:Given that SMA often involves fatty liver, dyslipidemia and insulin resistance, using the current mouse model, the authors could have explored the long-term effects of liver-specific depletion of SMN on metabolic phenotypes beyond P19, as well as systemic effects like glucose homeostasis. Given that the authors also report pancreatic cell defects, the long-term effect on insulin secretion and resistance could be further explored. The mechanistic link between a liver-specific SMN depletion and apparent pancreatic cell defects is also unclear.

We extended our analysis to include P60 mice and performed both liver and pancreatic analyses at this time point to address this suggestion. In addition, we discussed the liver-pancreas axis in the Discussion.

Discussion:This current work explores a novel mouse cross in order to specifically deplete liver SMN using an Albumin-Cre driver line. This provides insight into the contribution of liver-specific SMN protein to the pathology of SMA, which is relevant for understanding metabolic perturbations in SMA patients. Nonetheless, given that SMA in patients involve a systemic deletion or mutation of the SMN gene, the authors could emphasize the utility of this liver-specific mouse model, as opposed to using in vitro models, which have been recently reported (Leow et al, 2024, JCI). Authors should also discuss why a mild metabolic phenotype is observed in this current mouse model, as opposed to other SMA mouse models described in literature.

We appreciate the reviewer’s insightful comment. We have thoroughly addressed this suggestion in the Discussion section, particularly in lines 284-298; 309-322 and 334-359.

**Recommendations for the authors:**

**Reviewer #1 (Recommendations for the authors):**
(1) Longitudinal Studies: Conducting studies at maybe one more time points postnatally to provide a clearer picture of how liver-specific SMN depletion affects tissue pathology over time.

We extended our analysis to include P60 mice and performed both liver and pancreatic analyses at this time point to address this suggestion.

(2) Functional Assays: Incorporate glucose tolerance tests, insulin sensitivity tests, and more detailed metabolic profiling to better understand the physiological consequences of liver-specific SMN depletion on glucose metabolism and pancreatic function.

We sincerely thank the reviewer for this suggestion. We have included a full panel of metabolic hormones associated with glucose metabolism from animals at P19 and P60. These new data, along with additional figures, have now been provided in our revised manuscript.

(3) Mechanism: Discuss the molecular pathways affected by SMN depletion in the liver and pancreas. Mechanistic studies including transcriptomic or proteomic analyses to identify dysregulated pathways will help.

We appreciate the reviewer’s insightful comment. We have thoroughly addressed this suggestion in the Discussion section, particularly in lines 284-298 and 334-359.

(4) Typos in the abstract: beta cells secret insulin and alpha cells produce gulcagon.

Thank you for catching this error. It has been corrected to reflect that insulin is produced by beta cells and glucagon by alpha cells.

(5) Efficiency and specificity of the Alb-Cre: if possible, cross the Alb-Cre with the Rosa26 reporter line to test the efficiency and specificity of the Alb-Cre.

We agree that this would provide valuable insights. However, initiating a new breeding program to generate the required genotypes would take over a year and is beyond the scope of this study. To address this in part, we performed Cre immunostaining of the liver, pancreas, and spinal cord at P19, as well as the liver at P60. These results, now included in Supplemental Figure 1, demonstrate liver-specific expression and variability across hepatocytes.

**Reviewer #2 (Recommendations for the authors):**
The title of this manuscript is potentially misleading. The manuscript largely investigates the involvement of SMN protein on peripheral organs such as the liver, muscles, neuromuscular junction, and the pancreas. Yet, the title could be interpreted that the peripheral nervous system or central nervous system is the main focus. The title should be edited to indicate key terms such as "motor neuron and peripheral tissue pathology".

Thank you for pointing this out. We have revised the title to better represent the study’s focus. It is now “Impact of liver-specific survival motor neuron (SMN) depletion on central nervous system and peripheral tissue pathology”.

Suggestions:Please clarify and explain clearly the various mouse lines (Smn2B/+, Smn2B/- and +/+; Smn2B/F7) used as controls as the nomenclature used is confusing. In addition, authors could consider the use of a wild-type mouse line to be used as a control to validate changes in AlbCre/+; Smn2B/F7 mice.

We have now provided clarification on mouse line nomenclature in the Results section (lines 104–124). Full-body heterozygous mice (_Smn_2B/+) are used as controls due to their slightly reduced SMN protein levels and absence of phenotypic changes compared to wild-type mice.

Given that the main phenotype implicated by the liver-specific depletion of SMN protein in AlbCre/+;Smn2B/F7 mice is pancreatic abnormalities (changes in alpha- and beta- cell numbers and blood glucose levels), authors should expand further on the pancreatic phenotype.

We added a full panel of metabolic hormones related to glucose metabolism in animals at P19 and P60. Furthermore, this has been discussed in detail in lines 284-298 and 334-344 of the Discussion.

A pancreas-specific depletion of SMN would provide this current manuscript with a better understanding of the role of SMN in regulating SMA pathology and provide more definitive conclusions on the contribution of liver-specific SMN depletion on normal pancreatic function.

We agree that this would be very informative. However, to do this would require initiation of a new breeding program that will take more than a year to arrive at the right genotypes. Although valuable, it is beyond the scope of the present study.

The authors should also delineate the role of hepatic SMN in pancreatic function, and how the intrinsic liver-specific loss of SMN directly impacts the pancreas. Currently, literature demonstrates that the fatty liver phenotype in SMA patients is a primary SMN-dependent hepatocyte-intrinsic liver defect associated with mitochondrial and other hepatic metabolism implications (see Leow et al, 2024 J Clin Invest). Given that the authors describe that SMN protein levels are not altered in the pancreas of AlbCre/+;Smn2B/F7 mice at P19, the authors ought to clarify how pancreas development and function is impacted in this mouse model, whether in-utero or postnatally. This could potentially underscore the cross-talk between liver SMN and pancreas function.

We have discussed the relationship between hepatic SMN and pancreatic function in the Discussion at lines 284-298 and 334-359.

Authors should also perform some metabolic tolerance tests to both oral glucose and insulin at an older age (e.g. P60) to study their homeostasis in these mice. These would help to substantiate the authors' conclusion and provide the paper with a greater level of novelty.

We thank the reviewer for this suggestion. A full panel of metabolic hormones related to glucose metabolism at P19 and P60 has been included, supported by additional figures that enhance the manuscript's novelty and depth.

Authors mentioned in the Discussion in lines 238 to 240: "Altogether, our findings underscore the necessity of conducting further investigations at later time points to unveil potential modifications in other pathways and their repercussions on liver physiology". Please elucidate the effects of longer term liver-specific depletion of SMN beyond P19, such as the onset of NAFLD or a diabetic phenotype due to pancreatic dysfunctions.

We extended our data to include P60 mice and performed liver and pancreatic analyses at these time points. The observed effects were transient, possibly due to the stochastic nature of Cre expression.

In addition, while AlbCre/+;Smn2B/F7 mice had similar weight gain trends as controls, it does appear that AlbCre/+;Smn2B/F7 mice weigh more than their controls by P60 (Figure 9C). This data would provide more convincing evidence of the metabolic defects observed in these mice.

As per the reviewer’s suggestion, we included new data (Figure 9D) showing % weight gain at P60 normalized to basal weight at P7. However, no statistically significant differences were detected.

Other than protein quantification, authors should perform immunohistochemistry or in-situ hybridization of SMN and imaging of AlbCre/+;Smn2B/F7 organs to validate the loss of liver-specific SMN. It is unclear from western blots that the expression of SMN is only in hepatocytes.

We thank the reviewer for the suggestion. Unfortunately, SMN antibodies have not produced reliable tissue immunostaining. To address this, we performed Cre immunostaining of the liver, pancreas, and spinal cord at P19, and the liver at P60, which demonstrated liver-specific expression. These results are now included in Supplemental Figure 1.

Authors should consider re-wording lines 228 through 231: "While our current analysis did not reveal significant differences in AlbCre/+;Smn2B/F7 mice, the observed upward trend in transferrin and HO levels suggests ongoing changes in iron metabolism, which may not be fully manifested at P19". Alternatively, a higher number of mouse samples would allow them to qualify this statement. Authors should also consider comparing levels of liver biomarkers such as ALT and AST, to check for liver homeostatic function.

We have removed speculative statements to avoid unsupported claims.

Recommendations:The methods and additional details to generate the AlbCre/+;Smn2B/F7 should be explained better in section 2.1 of the Results. It is potentially confusing as to why these mice had to carry both 2B and F7 alleles. Additionally, the role of the F7 allele is not deliberately clear in the Introduction.

Additional details are now included in the Introduction (lines 87-90) and the Results section (lines 104-124).

Authors should refer to Leow et al 2024 (J Clin Invest) and discuss how their current findings compare with their hepatocyte-intrinsic SMN deficiency IPSCs model.We note a previous publication (Deguise et al 2021 Cell Mol Gastroenterol Hepatol) by the authors which characterized the Smn2B/- mouse model and its NAFLD/NASH features. From our understanding, the Smn2B/- mouse model appears to recapitulate SMA phenotype well, such as the early onset of hepatic steatosis and neurological conditions. As a follow-up to this publication, authors should discuss why this current study of a liver-specific SMN depletion is important and relevant to the study of SMA pathology.

We thank the reviewer for the insightful suggestions. We have included a discussion of these findings and their relevance to the study of SMA pathology in lines 284-298 and 309-322.

Minor corrections:Abstract (line 32) reads: "a decrease in insulin producing alpha-cells and an increase in glucagon producing beta-cells". The authors should clarify and correct as insulin producing beta-cells and glucagon producing alpha-cells.

Thank you for catching the error. We corrected the description of insulin- and glucagon-producing cells.

Please clarify the number and gender of mice used for weight tracking and motor function experiments up to P60 (Figure 9C). It would be inappropriate if male and female mice were plotted together. If so, authors should stratify data by gender.

We thank the reviewer for the suggestion. Unfortunately, we did not stratify the animals by sex due to the unequal and insufficient number of males and females in our study. To address this, we normalized weight gain to each animal’s starting weight, and no significant differences were observed (now shown in Figure 9D).

The number of figures should be reduced. We recommend merging Figures 1 and 2 (generation of AlbCre/+;Smn2B/F7 mouse line and validation) and Figures 3 and 4 (liver function). Figures 5 through 9 may be supplemental figures instead.

We thank the reviewer for the suggestions. We merged Figures 1 and 2, and Figures 3 and 4, as requested. However, we would prefer to keep the other figures within the main results as they assess the impact of liver-specific depletion of SMN on other pathologies within the mouse model.

Standardize the use of asterisks and reporting p-values in Figure 2. All other figures in the manuscript utilize asterisks, but Figures 2C', 2D' and 2E' use p-values across comparisons.

P-values were included only when they approached statistical significance, providing additional clarity to the results.

It is unclear what the white arrow in Figure 7A indicates.

It is meant to point out the absence of an innervating axon. Please see Figure 5 legend, lines 801-802.

Note spelling errors in Figures 8B and 8C: 'Muscle flber'.

Thank you for catching this. We have corrected the typo to indicate muscle fiber instead.

Please clarify if muscle fiber size should be indicated as µm2 instead of µ2 in Figures 8B and 8C, as written in Materials and Methods under line 394.

Thank you for catching this. We corrected the typo to indicate µm2 instead.